# Pseudo-Calibration: Improving Predictive Uncertainty Estimation in Domain Adaptation

## Abstract

Unsupervised domain adaptation (UDA) has seen significant efforts to enhance model accuracy for an unlabeled target domain with the help of one or more labeled source domains. However, UDA models often exhibit poorly calibrated predictive uncertainty on target data, a problem that remains under-explored and poses risks in safety-critical UDA applications. The two primary challenges in addressing predictive uncertainty calibration in UDA are the absence of labeled target data and severe distribution shifts between the two domains. Traditional supervised calibration methods like *temperature scaling* are inapplicable due to the former challenge. Recent studies address the first challenge by employing *importance-weighting* with labeled source data but still suffer from the second challenge and require additional complex density modeling. We propose Pseudo-Calibration (PseudoCal), a novel post-hoc calibration framework. Unlike prior approaches, we consider UDA calibration as a target-domain specific unsupervised problem rather than a *covariate shift* problem across domains. Our innovative use of inference-stage *mixup* and *cluster assumption* guarantees that a synthesized labeled pseudo-target set captures the structure of the real unlabeled target data. In this way, we turn the unsupervised calibration problem into a supervised one, readily solvable with *temperature scaling*. Extensive empirical evaluation across 5 diverse UDA scenarios involving 10 UDA methods consistently demonstrates the superior performance of PseudoCal over alternative calibration methods.

## 1 Introduction

In recent years, unsupervised domain adaptation (UDA) (Pan & Yang, 2009) has gained popularity for enhancing the generalization of deep learning models (He et al., 2016; Dosovitskiy et al., 2021) from labeled source domains to an unlabeled target domain that share similar tasks but have varying data distributions. Notable progress has been achieved in developing effective UDA methods (Ganin & Lempitsky, 2015; Long et al., 2018; Saito et al., 2018), practical applications (Chen et al., 2018; Tsai et al., 2018), and real-world settings (Long et al., 2015; Cao et al., 2018; Liang et al., 2020a), with a predominant focus on improving target domain model accuracy.

However, for a classification model, achieving reliable predictive uncertainty estimation is as crucial as high accuracy, especially in safety-critical decision-making scenarios like autonomous driving and medical diagnosis (Guo et al., 2017). Calibrated models should produce probability predictions that accurately reflect correctness likelihood (Guo et al., 2017; Lakshminarayanan et al., 2017). Although predictive uncertainty calibration has garnered substantial attention in IID supervised learning tasks with deep models (Thulasidasan et al., 2019; Krishnan & Tickoo, 2020), the calibration problem in UDA remained largely unexplored until a pioneering UDA study (Wang et al., 2020), which revealed that improved UDA model accuracy comes at the expense of poor uncertainty calibration on target data. This phenomenon is vividly illustrated in Figure 1(a), where increasing target data accuracy is accompanied by significant overfitting of the negative log-likelihood (NLL) loss during adaptation. Calibrating predictive uncertainty in UDA presents unique challenges compared with the IID situation. The first challenge is the absence of labeled data in the target domain, rendering the direct application of supervised IID calibration methods like *temperature scaling* (Guo et al., 2017) impossible. Another significant challenge arises from severe domain distribution shifts between source and target. Consequently, UDA models calibrated with labeled source data cannot ensure effective calibration for unlabeled data in the target domain (Wang et al., 2020).

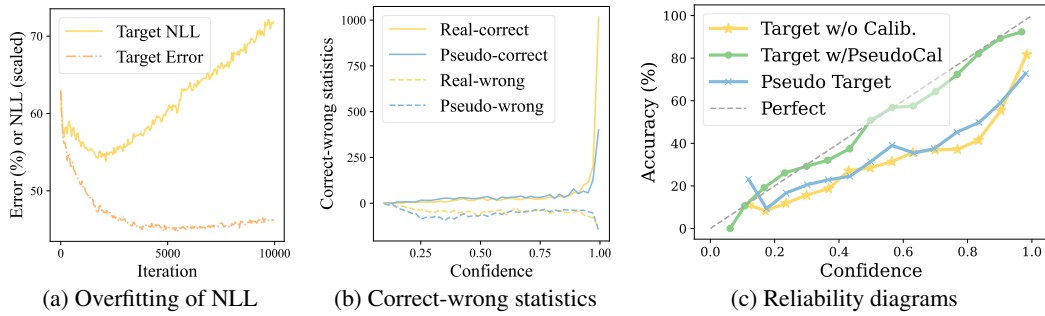

(a) Overfitting of NLL     (b) Correct-wrong statistics     (c) Reliability diagrams

Figure 1: ATDOC (Liang et al., 2021) on a closed-set UDA task Ar → Cl. (a) illustrates the target error and target NLL loss (rescaled to match error) during UDA training. (b) divides confidence values into 50 bins, displaying the count of correct and wrong predictions in each bin. For real target data, correctness is determined by comparing predictions with ground truths, and for pseudo-target data, it's assessed by comparing predictions with synthesized labels. (c) shows reliability diagrams (Guo et al., 2017) for both pseudo and real target data. Perfect: ideal predictions without miscalibrations.

To address these challenges, existing approaches (Park et al., 2020; Pampari & Ermon, 2020; Wang et al., 2020) treat calibration in UDA as a *covariate shift* problem (Sugiyama et al., 2007) across domains. They typically employ *importance weighting* (Cortes et al., 2008) to estimate weights for source samples based on the similarity to target data and then perform sample-weighted *temperature scaling* with a labeled source validation set. However, these methods have some drawbacks that impede effective and efficient model calibration in UDA. Firstly, *importance weighting* may not be reliable under severe *covariate shift* and other distribution shifts, such as label shift (Park et al., 2020). Secondly, despite being based on the simple and post-hoc *temperature scaling*, all of these approaches require additional model training for accurate density estimation, adding complexity. Lastly, these methods rely on labeled source data, which limits their applicability in privacy-preserving UDA scenarios like the recent source-free UDA settings (Li et al., 2020; Liang et al., 2020a; 2022).

In contrast, we adopt a novel perspective, treating UDA calibration as an unsupervised calibration problem specific to the target domain, which allows us to focus solely on the first challenge: the absence of labeled target data for supervised calibration. We first study the 'Oracle' case of using labeled target data for *temperature scaling* and then factorize its NLL objective into a joint optimization involving both correct and wrong predictions. This factorization uncovers a key insight with *temperature scaling*: datasets with similar correct-wrong statistics should share similar temperatures. Inspired by this insight, we introduce a novel post-hoc calibration framework called Pseudo-Calibration (PseudoCal). PseudoCal is based on *temperature scaling* and estimates the temperature for target data through calibration on a synthesized labeled pseudo-target dataset that mimics similar correct-wrong statistics as the real target data. Concretely, PseudoCal utilizes *mixup* (Zhang et al., 2018) during the inference stage with unlabeled target data to generate a labeled pseudo-target set. It then performs supervised calibration on this labeled set to determine the final temperature. PseudoCal's effectiveness depends on the presence of similar correct-wrong statistics between pseudo and real target data. We elucidate the behind support with an intuitive analysis grounded in the well-established *cluster assumption* (Grandvalet & Bengio, 2004). UDA models adhering to this assumption can ensure sample-level correspondence between each pseudo-target sample and its primary real target sample used in the *mixup* operation. Specifically, pseudo-target samples with correct predictions correspond to correct real target samples, and vice versa, as shown in Figure 1(b). Benefitting from the high resemblance of correct-wrong statistics between our synthesized pseudo-target and real target, PseudoCal achieves significantly improved calibration performance, as demonstrated in Figure 1(c).

We make three key contributions: 1) We explore the under-studied problem of predictive uncertainty calibration in UDA from a novel target-domain perspective, enabling a unified approach across diverse UDA scenarios, including those with label shift or limited source access. 2) We propose a novel calibration framework, PseudoCal, which only requires unlabeled target data and a fixed UDA model. PseudoCal adopts inference-stage *mixup* to synthesize a labeled pseudo-target set, successfully converting the challenging unsupervised calibration problem into a readily solvable supervised one. 3) We conduct a comprehensive evaluation of PseudoCal, involving 5 calibration baselines, to calibrate 10 UDA methods across 5 UDA scenarios. Experimental results demonstrate that, on average, PseudoCal consistently and significantly outperforms all other calibration methods.

Table 1: Comparisons of typical methods for predictive uncertainty calibration in UDA.

| Calibration Method | Covariate Shift | Label Shift | No harm to accuracy | No extra training | No source data |
|---|---|---|---|---|---|
| TempScal-src (Guo et al., 2017) | ✗ | ✗ | ✓ | ✓ | ✗ |
| MC-Dropout (Gal & Ghahramani, 2016) | ✓ | ✓ | ✗ | ✓ | ✓ |
| Ensemble (Lakshminarayanan et al., 2017) | ✓ | ✓ | ✓ | ✗ | ✓ |
| CPCS (Park et al., 2020) | ✓ | ✗ | ✓ | ✗ | ✗ |
| TransCal (Wang et al., 2020) | ✓ | ✗ | ✓ | ✗ | ✗ |
| PseudoCal (Ours) | ✓ | ✓ | ✓ | ✓ | ✓ |

## 2 RELATED WORK

**Unsupervised domain adaptation (UDA)** has been extensively studied in image classification tasks. Mainstream methods can be categorized into two lines: 1) Distribution alignment across domains using specific discrepancy measures (Long et al., 2015; Sun & Saenko, 2016) or adversarial learning (Ganin & Lempitsky, 2015; Tzeng et al., 2017; Long et al., 2018; Saito et al., 2018), and 2) Target domain-based learning with self-training (Shu et al., 2018; Liang et al., 2021) or regularizations (Xu et al., 2019; Cui et al., 2020; Jin et al., 2020). Moreover, UDA has also been studied in object detection (Chen et al., 2018; Saito et al., 2018) and image segmentation (Tsai et al., 2018; Vu et al., 2019). Initially, UDA was based on the *covariate shift* assumption (Sugiyama et al., 2007) – two domains share similar label and conditional distributions but have different input distributions. This is commonly referred to as closed-set UDA. In recent years, new practical settings have arisen, notably addressing label shift (Lipton et al., 2018). These include partial-set UDA (Cao et al., 2018; Liang et al., 2020b), where some source classes are absent in the target domain, and open-set UDA (Panareda Busto & Gall, 2017), where the target domain includes samples from unknown classes. Recently, there has been a growing interest in a setting called source-free UDA, which can preserve source privacy. Source-free UDA has two key settings: the white-box setting (Li et al., 2020; Liang et al., 2020a) uses the source model for target adaptation, while the stricter black-box setting (Zhang et al., 2021; Liang et al., 2022) only employs the source model for inference.

**Predictive uncertainty calibration** was initially stuidied on binary classification tasks (Zadrozny & Elkan, 2001; 2002; Platt et al., 1999). Guo et al. (2017) extends *Platt scaling* (Platt et al., 1999) to multi-class classification and introduces *matrix scaling* (MatrixScal), *vector scaling* (VectorScal), and *temperature scaling* (TempScal). These post-hoc methods require a labeled validation set for calibration. On the other hand, there are methods that address calibration during model training, including *Monte Carlo Dropout* (MC-Dropout)(Gal & Ghahramani, 2016), Ensemble (Lakshminarayanan et al., 2017), and *Stochastic Variational Bayesian Inference* (SVI) (Blundell et al., 2015; Louizos & Welling, 2017; Wen et al., 2018). However, an evaluation in (Ovadia et al., 2019) reveals that these methods do not maintain calibration performance under dataset shift. In addition, there is growing interest in calibration under distribution shifts (Alexandari et al., 2020; Wang et al., 2020; Park et al., 2020) and in semantic segmentation tasks (Ding et al., 2021; Wang et al., 2022; de Jorge et al., 2023). In this paper, we specifically address the calibration problem in single-source UDA. A vanilla baseline is to apply IID calibration methods such as TempScal with a labeled source validation set, dubbed TempScal-src. Regarding calibration methods considering the domain distribution shifts, the mainstream idea is to utilize *importance weighting* (Cortes et al., 2008) to address calibration under *covariate shift* in UDA, exemplified by CPCS (Park et al., 2020) and TransCal (Wang et al., 2020). Some works perturb the source validation set to serve as a general target set (Tomani et al., 2021; Salvador et al., 2021) or employ it for density estimation (Tomani et al., 2023). More recently, some methods (Gong et al., 2021; Yu et al., 2022) have utilized multiple source domains to calibrate the unlabeled target domain in UDA. Additionally, there are training-stage calibration methods that employ smoothed labels (Thulasidasan et al., 2019; Liu et al., 2022) or optimize accuracy-uncertainty differentiably (Krishnan & Tickoo, 2020). Among these methods, CPCS and TransCal are noteworthy as they specifically address transductive target calibration in UDA. For more general approaches like MC-Dropout and Ensemble, we compare our method directly with Ensemble because it consistently outperforms MC-Dropout. Table 1 presents a comprehensive comparison of these typical UDA calibration methods. In contrast to existing calibration methods, PseudoCal stands out by not requiring any extra model training. It is a simple, post-hoc, and general calibration approach, solely relying on a fixed or even black-box UDA model and unlabeled target data for calibration.

## 3 APPROACH

We begin by introducing unsupervised domain adaptation (UDA) with a $C$-way image classification task. UDA generally involves two domains: a labeled source domain and an unlabeled target domain. The source domain $\mathcal{D}_{\mathrm{s}} = \{(x_{\mathrm{s}}^i, y_{\mathrm{s}}^i)\}_{i=1}^{n_{\mathrm{s}}}$ consists of $n_{\mathrm{s}}$ images $x_{\mathrm{s}}$ with their corresponding labels $y_{\mathrm{s}}$, where $x_{\mathrm{s}}^i \in \mathcal{X}_{\mathrm{s}}$ and $y_{\mathrm{s}}^i \in \mathcal{Y}_{\mathrm{s}}$. The target domain $\mathcal{D}_{\mathrm{t}} = \{x_{\mathrm{t}}^i\}_{i=1}^{n_{\mathrm{t}}}$ contains unlabeled images $x_{\mathrm{t}}$, where $x_{\mathrm{t}}^i \in \mathcal{X}_{\mathrm{t}}$. The objective of UDA is to learn a UDA model $\phi$ that can predict the unknown ground truth labels $\{y_{\mathrm{t}}^i\}_{i=1}^{n_{\mathrm{t}}}$ for the target domain, utilizing data from both domains simultaneously (Ganin & Lempitsky, 2015) or sequentially (Liang et al., 2020a).

Next, we introduce predictive uncertainty calibration and relevant metrics. When feeding an arbitrary sample $(x, y)$ into the UDA model $\phi$, we can obtain the predicted class $\hat{y}$ and the corresponding softmax-based confidence $\hat{p}$. Ideally, the confidence should accurately reflect the probability of correctness, expressed as $\mathbb{P}(\hat{y} = y | \hat{p} = p) = p, \ \forall \, p \in [0, 1]$. This perfect calibration, also known as *Perfect*, is impossible to achieve (Guo et al., 2017). The widely used metric for evaluating calibration error is the expected calibration error (ECE) (Guo et al., 2017). ECE involves partitioning probability predictions into $M$ bins, with $B_m$ representing the indices of samples falling into the $m$-th bin. It calculates the weighted average of the accuracy-confidence difference across all bins:

$$\mathcal{L}_{\mathrm{ECE}} = \sum_{m=1}^{M} \frac{|B_m|}{n} |\mathrm{acc}(\mathrm{B_m}) - \mathrm{conf}(\mathrm{B_m})|$$

Here, $n$ represents the number of samples, and for the $m$-th bin, the accuracy is computed as $\mathrm{acc}\,(B_m) = |B_m|^{-1} \sum_{i \in B_m} \mathbb{1}(\hat{y}_i = y_i)$, and the confidence is computed as $\mathrm{conf}\,(B_m) = |B_m|^{-1} \sum_{i \in B_m} \hat{p}_i$. The introduction of additional popular metrics, such as NLL (Goodfellow et al., 2016) and Brier Score (BS) (Brier et al., 1950), is provided in Appendix B for further reference.

### 3.1 SUPERVISED 'ORACLE': FACTORIZED TEMPERATURE SCALING

Unlike the mainstream cross-domain *covariate shift* perspective, we view calibration in UDA as an unsupervised calibration problem within the unlabeled target domain. Before tackling this challenging problem, we study an 'Oracle' solution based on supervised *temperature scaling* (TempScal) (Guo et al., 2017). TempScal is a post-hoc calibration method that optimizes a temperature scalar, denoted as $T$, on a labeled validation set using the NLL loss between the temperature-flattened softmax predictions and the ground truth labels. For the unlabeled target domain in UDA, we define the calibration achieved by applying TempScal with raw target predictions and unattainable target ground truths as the 'Oracle' calibration. Let $z$ represent the corresponding logit vector for the image input $x$, and let $\sigma(\cdot)$ denote the softmax function. The 'Oracle' target temperature, denoted as $T_{\mathrm{o}}$, can be obtained using the original *temperature scaling* optimization formulated as follows.

$$T_{\mathrm{o}} = \arg\min_{T} \ \mathbb{E}_{(x_{\mathrm{t}}^i, y_{\mathrm{t}}^i) \in \mathcal{D}_{\mathrm{t}}} \ \mathcal{L}_{\mathrm{NLL}} \left( \sigma(z_{\mathrm{t}}^i / T), y_{\mathrm{t}}^i \right) \tag{1}$$

With further analysis, we observe that target samples can be classified as either being correctly or wrongly predicted when evaluated by target ground truths. Moreover, both types of samples have contrasting effects on the temperature optimization process. Specifically, the NLL minimization favors a small temperature to sharpen the correct predictions and a large temperature to flatten the wrong predictions. Therefore, we can factorize Equation 1 as follows:

$$T_{\mathrm{o}} = \arg\min_{T} \ \frac{N_{\mathrm{c}}}{N} \mathbb{E}_{(x_{\mathrm{t}}^i, y_{\mathrm{t}}^i) \in \mathcal{D}_{\mathrm{c}}} \ \mathcal{L}_{\mathrm{NLL}} \left( \sigma(z_{\mathrm{t}}^i / T), y_{\mathrm{t}}^i \right) + \frac{N_{\mathrm{w}}}{N} \mathbb{E}_{(x_{\mathrm{t}}^j, y_{\mathrm{t}}^j) \in \mathcal{D}_{\mathrm{w}}} \ \mathcal{L}_{\mathrm{NLL}} \left( \sigma(z_{\mathrm{t}}^j / T), y_{\mathrm{t}}^j \right), \tag{2}$$

where $\mathcal{D}_{\mathrm{c}}$ represents the set of correctly predicted target samples, comprising $N_{\mathrm{c}}$ instances. Similarly, $\mathcal{D}_{\mathrm{w}}$ denotes the set of wrongly predicted target samples, consisting of $N_{\mathrm{w}}$ instances. Obviously, this factorization suggests that when applying TempScal to another labeled set with matching correct-wrong statistics (i.e., the same count of correct and wrong predictions) as the 'Oracle' calibration in Equation 2, the objective of the NLL optimization remains highly consistent, yielding a temperature approximation close to the target oracle temperature $T_{\mathrm{o}}$.

### 3.2 UNSUPERVISED SOLUTION: PSEUDO-CALIBRATION

Inspired by this factorization, we introduce our Pseudo-Calibration (PseudoCal) framework. The main idea is to use the unlabeled target data to synthesize a labeled pseudo-target set that mimics the correct-wrong statistics of the real target set and then apply TempScal to this labeled set.

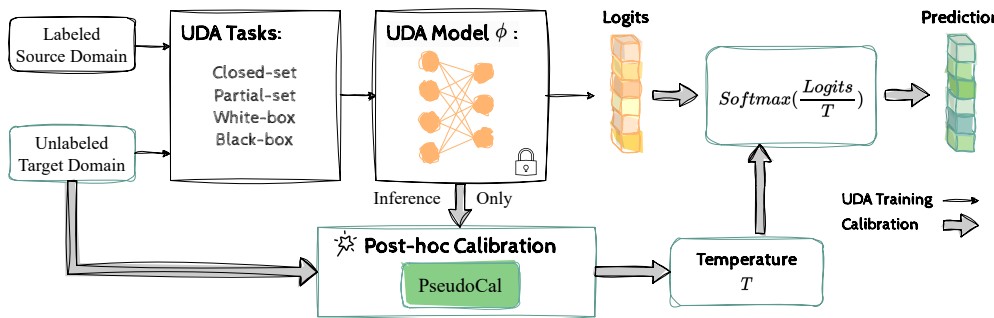

Figure 2: The pipeline of PseudoCal for calibrating predictive uncertainty in UDA.

With only unlabeled target data and a fixed UDA model, the use of predicted labels as pseudo-labels (Lee et al., 2013) is a simple method to generate a labeled set. However, optimizing NLL between raw target predictions and pseudo-labels treats all predictions as correct, ignoring the optimization of wrong predictions in Equation 2. This mismatch in correct-wrong statistics can result in poor calibration performance, as demonstrated in Table 9. Instead, we employ *mixup* (Zhang et al., 2018) with data across clusters (i.e., with different pseudo-labels), generating mixed samples that inherently include both correct and wrong predictions when evaluated with mixed labels.

**Step 1: Pseudo-target synthesis.** We generate a pseudo-target set by applying *mixup* to target samples in the inference stage. Specifically, a pseudo-target sample $x_{\mathrm{pt}}$ and its label $y_{\mathrm{pt}}$ are obtained by taking a convex combination of a pair of real target samples $x_{\mathrm{t}}^i, x_{\mathrm{t}}^j$ from different clusters and their pseudo-labels $\hat{y}_{\mathrm{t}}^i, \hat{y}_{\mathrm{t}}^j$. Consequently, we obtain a labeled pseudo-target set $\{(x_{\mathrm{pt}}^i, y_{\mathrm{pt}}^i)\}_{i=1}^{n_{\mathrm{pt}}}$, where $n_{\mathrm{pt}}$ represents the amount. The general process of pseudo-target synthesis is formulated as follows:

$$x_{\mathrm{pt}} = \lambda * x_{\mathrm{t}}^i + (1 - \lambda) * x_{\mathrm{t}}^j, \qquad y_{\mathrm{pt}} = \lambda * \hat{y}_{\mathrm{t}}^i + (1 - \lambda) * \hat{y}_{\mathrm{t}}^j, \tag{3}$$

where $\lambda \in (0.5, 1.0)$ is a fixed scalar used as the mix ratio, different from that in common *mixup*.

**Step 2: Supervised calibration.** Using the synthesized labeled pseudo-target set $\{(x_{\mathrm{pt}}^i, y_{\mathrm{pt}}^i)\}_{i=1}^{n_{\mathrm{pt}}}$, we can easily determine the optimal pseudo-target temperature through TempScal. This estimated temperature serves as an approximation of the 'Oracle' target temperature $T_{\mathrm{o}}$.

With the above two simple steps, PseudoCal successfully transforms the challenging unsupervised calibration problem associated with the unlabeled real target set into a supervised one with the labeled pseudo-target set and readily solves it with TempScal. The pipeline of PseudoCal is illustrated in Figure 2, where the UDA model is utilized as a black box solely for inferring the predictions of input data. A comprehensive Pytorch-style pseudocode for PseudoCal is provided in Appendix A.

**Analysis.** Built upon the well-established *cluster assumption* (Grandvalet & Bengio, 2004; Chapelle & Zien, 2005), we intuitively analyze how mixed samples can exhibit similar correct-wrong statistics as real target data, as empirically depicted in Figure 1(b). This assumption suggests that within a well-learned data structure, samples located far from the classification boundary are more likely to be correctly classified, while those near the boundary are prone to misclassification. While previous works often incorporate this assumption as an objective in model training (Shu et al., 2018; Verma et al., 2022), our focus here is to employ it for explaining the inference behavior of a UDA model $\phi$. We assume that the model has effectively learned the underlying target-domain structure. For simplicity, let's assume all involved labels in Equation 3 are one-hot, and consider a fixed mix ratio $\lambda$ noticeably greater than 0.5 (e.g., 0.65). This ensures a clear distinction between two involved real samples: one primary sample $x_{\mathrm{t}}^i$ with a mix ratio greater than 0.5, determining the mixed label $y_{\mathrm{pt}}$ for the mixed sample $x_{\mathrm{pt}}$, and the other as the minor sample $x_{\mathrm{t}}^j$, serving only as an input perturbation. If $x_{\mathrm{pt}}$ yields a correct model prediction $\hat{y}_{\mathrm{pt}}$ evaluated by its mixed label (i.e., $\hat{y}_{\mathrm{pt}} == y_{\mathrm{pt}}$), it suggests that the real sample $x_{\mathrm{t}}^i$ maintains its prediction after cross-cluster perturbation. This implies that $x_{\mathrm{t}}^i$ is likely distant from the classification boundary, and its prediction or pseudo-label $\hat{y}_{\mathrm{t}}^i$ is genuinely correct when evaluated against its ground truth $y_{\mathrm{t}}^i$. Similarly, if $x_{\mathrm{pt}}$ yields a wrong model prediction $\hat{y}_{\mathrm{pt}}$ (i.e., $\hat{y}_{\mathrm{pt}} \neq y_{\mathrm{pt}}$), we can reasonably infer that $x_{\mathrm{t}}^i$ has a truly incorrect prediction. The presence of sample-level correspondence, when observed at the dataset level, manifests as similar correct-wrong statistics. However, this correspondence may not hold under extreme perturbation degrees (i.e., $\lambda$ near 0.5 or 1.0). Kindly refer to Appendix D for detailed empirical evidence.

## 4 EXPERIMENTS

### 4.1 SETTINGS

**Datasets.** For image classification, we adopt 5 popular UDA benchmarks of varied scales. *Office-31* (Saenko et al., 2010) is a small-scale benchmark with 31 classes in 3 domains: Amazon (A), DSLR (D), and Webcam (W). *Office-Home* (Venkateswara et al., 2017) is a medium-scale benchmark with 65 classes in 4 domains: Art (Ar), Clipart (Cl), Product (Pr), and Real-World (Re). *VisDA* (Peng et al., 2017) is a large-scale benchmark with over 200k images across 12 classes in 2 domains: Training (T) and Validation (V). *DomainNet* (Peng et al., 2019) is a large-scale benchmark with 600k images. We take a subset of 126 classes with 7 tasks(Saito et al., 2019) from 4 domains: Real (R), Clipart (C), Painting (P), and Sketch (S). *Image-Sketch* (Wang et al., 2019) is a large-scale benchmark with 1000 classes in 2 domains: ImageNet (I) and Sketch (S). For semantic segmentation, we use *Cityscapes*(Cordts et al., 2016) as the target domain and either *GTA5*(Richter et al., 2016) or *SYNTHIA* (Ros et al., 2016) as the source.

**UDA methods.** We evaluate calibration on 10 UDA methods across 5 UDA scenarios. For image classification, we cover closed-set UDA methods (ATDOC (Liang et al., 2021), BNM (Cui et al., 2020), MCC (Jin et al., 2020), CDAN (Long et al., 2018), SAFN (Xu et al., 2019), MCD (Saito et al., 2018)), partial-set UDA methods (ATDOC (Liang et al., 2021), MCC (Jin et al., 2020), PADA (Cao et al., 2018)), the whit-box source-free UDA method (SHOT (Liang et al., 2020a)), and the black-box source-free UDA method (DINE (Liang et al., 2022)). For semantic segmentation, we focus on calibrating source-only models without any adaptation.

**Calibration baselines.** For a comprehensive comparison, we consider 5 typical calibration baselines in UDA, including the no calibration baseline (No Calib.), source-domain calibration (TempScal-src (Guo et al., 2017)), cross-domain calibration (CPCS (Park et al., 2020), TransCal (Wang et al., 2020)), and a generic calibration method (Ensemble (Lakshminarayanan et al., 2017)).

**Implementation details.** We train all UDA models using their official code until convergence on a single RTX TITAN 16GB GPU. We adopt ResNet-101 (He et al., 2016) for *VisDA* and segmentation tasks, ResNet-34 (He et al., 2016) for *DomainNet*, and ResNet-50 (He et al., 2016) for all other tasks. For PseudoCal, a fixed mix ratio $\lambda$ of $0.65$ is employed in all experiments. The UDA model is fixed for only inference use. We use it for one-epoch inference with *mixup* to generate the labeled pseudo-target set. The reported results are averaged over five random runs.

Table 2: ECE (%) of closed-set UDA on *Office-Home* (*Home*). Lower is better. **bold**: Best case.

| Method | ATDOC | | | | | BNM | | | | | MCC | | | | |
| --- | →Ar | →Cl | →Pr | →Re | avg | →Ar | →Cl | →Pr | →Re | avg | →Ar | →Cl | →Pr | →Re | avg |
| No Calib. | 10.07 | 22.35 | 8.61 | 6.06 | 11.77 | 30.97 | 39.85 | 19.70 | 16.73 | 26.81 | 13.25 | 23.11 | 12.33 | 10.53 | 14.81 |
| TempScal-src | 6.19 | 17.54 | **3.98** | **3.03** | 7.68 | 23.11 | 30.32 | 13.70 | 10.25 | 19.35 | 6.74 | 16.25 | **5.08** | 4.10 | 8.04 |
| CPCS | 14.13 | 14.75 | 11.02 | 7.33 | 11.81 | 24.76 | 25.02 | 14.90 | 8.80 | 18.37 | 19.11 | 28.59 | 14.65 | 5.55 | 16.97 |
| TransCal | 18.09 | 6.52 | 16.03 | 18.29 | 14.73 | 17.44 | 27.22 | 9.14 | 5.47 | 14.82 | 11.73 | 3.86 | 6.70 | 8.16 | 7.61 |
| Ensemble | 7.38 | 18.01 | 5.51 | 4.22 | 8.78 | 22.50 | 30.68 | 14.38 | 12.53 | 20.02 | 9.76 | 19.20 | 9.48 | 7.90 | 11.58 |
| PseudoCal (ours) | **2.42** | **2.93** | 5.84 | 5.07 | **4.07** | **17.34** | **16.03** | **6.20** | **4.68** | **11.06** | **2.85** | **2.25** | 5.18 | **3.57** | **3.47** |
| Oracle | 1.71 | 1.91 | 2.29 | 1.69 | 1.90 | 2.20 | 2.53 | 2.36 | 1.60 | 2.17 | 2.25 | 1.64 | 2.22 | 1.91 | 2.00 |
| Accuracy (%) | 66.42 | 52.39 | 76.60 | 77.74 | 68.29 | 65.42 | 53.69 | 76.51 | 78.98 | 68.65 | 61.03 | 47.47 | 72.37 | 74.03 | 63.73 |

| Method | CDAN | | | | | SAFN | | | | | MCD | | | | | *Home* |
| --- | →Ar | →Cl | →Pr | →Re | avg | →Ar | →Cl | →Pr | →Re | avg | →Ar | →Cl | →Pr | →Re | avg | AVG |
| No Calib. | 13.38 | 22.94 | 12.15 | 10.00 | 14.62 | 16.57 | 27.90 | 13.16 | 11.93 | 17.39 | 16.36 | 25.96 | 13.29 | 11.97 | 16.89 | 17.05 |
| TempScal-src | 6.89 | 15.44 | 5.01 | 4.19 | 7.88 | 6.99 | 16.13 | 4.56 | **4.07** | 7.94 | 6.01 | 12.15 | **3.56** | 3.54 | 6.31 | 9.53 |
| CPCS | 18.38 | 33.56 | 15.29 | 9.90 | 19.28 | 14.98 | 30.54 | 10.06 | 12.11 | 16.92 | 25.13 | 27.26 | 10.17 | 14.29 | 19.21 | 17.09 |
| TransCal | 14.76 | 4.72 | 12.07 | 13.73 | 11.32 | 3.50 | 6.87 | **3.77** | 4.15 | 4.57 | 10.78 | **2.66** | 10.31 | 11.27 | 8.76 | 10.30 |
| Ensemble | 10.07 | 18.58 | 9.15 | 7.23 | 11.26 | 14.82 | 24.90 | 11.17 | 9.86 | 15.19 | 12.36 | 20.87 | 8.93 | 7.64 | 12.45 | 13.21 |
| PseudoCal (ours) | **5.10** | **3.72** | **4.71** | **2.40** | **3.98** | **3.05** | **3.34** | 4.66 | 4.37 | **4.41** | **4.07** | 2.86 | 6.26 | **3.72** | **4.23** | **5.20** |
| Oracle | 3.61 | 2.84 | 2.26 | 1.94 | 2.66 | 1.96 | 2.48 | 2.52 | 1.74 | 2.17 | 2.65 | 2.27 | 2.30 | 2.22 | 2.36 | 2.21 |
| Accuracy (%) | 62.26 | 49.99 | 71.19 | 73.79 | 64.31 | 65.84 | 51.90 | 73.78 | 75.09 | 66.66 | 59.04 | 46.80 | 68.75 | 71.39 | 61.49 | 65.52 |

### 4.2 RESULTS

We evaluate PseudoCal across 5 UDA scenarios. For classification, we report the averaged ECE across UDA tasks sharing the same target domain in Tables 2-6. For segmentation, we take each pixel as a sample and report the results in Table 7. 'Oracle' refers to the 'Oracle' calibration with target labels. 'Accuracy' (%) denotes the target accuracy of the fixed UDA model. Refer to Appendix C for segmentation details and Appendix F for full results, including ECE results for each UDA task.

**Closed-set UDA.** We evaluate 6 UDA methods on 4 benchmarks for closed-set UDA. Specifically, we report the ECE for *Office-Home* in Table 2, ECE for both *Office-31* and *VisDA* in Table 3, and ECE for *DomainNet* in Table 4. PseudoCal consistently achieves a low ECE close to 'Oracle', significantly outperforming other calibration methods by a large margin. On the evaluated benchmarks, PseudoCal shows average ECE improvements of $4.33\%$ on *Office-Home*, $1.88\%$ on *Office-31*, $2.77\%$ on *VisDA*, and $5.95\%$ on *DomainNet* when compared to the second-best calibration method.

Table 3: ECE (%) of closed-set UDA on *Office-31* (*Office*) and *VisDA*.

| Method | ATDOC | | | | | BNM | | | | | MCC | | | | |
|---|---|---|---|---|---|---|---|---|---|---|---|---|---|---|---|
| | →A | →D | →W | avg | T→V | →A | →D | →W | avg | T→V | →A | →D | →W | avg | T→V |
| No Calib. | 12.17 | 4.59 | 6.66 | 7.81 | 10.38 | 23.41 | 11.12 | 8.27 | 14.27 | 17.10 | 19.29 | 6.18 | 7.80 | 11.09 | 17.42 |
| TempScal-src | 22.39 | **3.39** | 4.18 | 9.99 | 10.53 | 23.85 | 9.23 | 4.98 | 12.69 | 13.72 | 21.38 | 3.79 | 3.00 | 9.39 | 13.28 |
| CPCS | 24.64 | 7.98 | 8.94 | 13.85 | 16.65 | 22.45 | 11.65 | **2.02** | 12.04 | 15.36 | 30.16 | 4.69 | 3.03 | 12.63 | 7.14 |
| TransCal | 12.14 | 14.21 | 14.64 | 13.67 | 6.36 | 14.86 | **5.22** | 2.70 | 7.59 | 8.79 | 6.53 | 3.77 | 3.91 | 4.74 | 12.21 |
| Ensemble | 9.79 | 3.60 | **4.09** | 5.83 | 8.53 | 19.77 | 6.92 | 4.63 | 10.44 | 14.84 | 17.48 | 3.07 | 4.88 | 8.48 | 15.32 |
| PseudoCal (ours) | **3.85** | 6.64 | 4.98 | **5.16** | **5.27** | **9.48** | 6.30 | 3.97 | **6.58** | **3.03** | **4.61** | **2.68** | **2.82** | **3.37** | **1.20** |
| Oracle | 2.13 | 2.49 | 3.15 | 2.59 | 0.52 | 2.52 | 2.65 | 1.40 | 2.19 | 0.93 | 2.24 | 2.36 | 2.67 | 2.42 | 1.12 |
| Accuracy (%) | 73.23 | 91.57 | 88.93 | 84.58 | 75.96 | 72.56 | 88.35 | 90.94 | 83.95 | 76.23 | 69.69 | 91.37 | 89.06 | 83.37 | 78.00 |

| Method | CDAN | | | | | SAFN | | | | | MCD | | | | | *Office* | *VisDA* |
|---|---|---|---|---|---|---|---|---|---|---|---|---|---|---|---|---|---|
| | →A | →D | →W | avg | T→V | →A | →D | →W | avg | T→V | →A | →D | →W | avg | T→V | AVG | AVG |
| No Calib. | 17.02 | 9.34 | 7.96 | 11.44 | 15.90 | 21.34 | 6.17 | 6.68 | 11.40 | 18.53 | 16.71 | 9.49 | 8.88 | 11.69 | 17.58 | 11.28 | 16.15 |
| TempScal-src | 18.54 | 5.70 | 3.41 | 9.21 | 14.19 | 23.95 | 3.21 | 2.83 | 9.99 | 14.40 | 25.37 | 3.44 | **2.36** | 10.39 | 10.22 | 10.28 | 12.72 |
| CPCS | 17.47 | 30.95 | 5.67 | 18.03 | 15.45 | 23.15 | 8.21 | 18.21 | 16.52 | 17.88 | 27.69 | 11.85 | 19.01 | 19.52 | 10.56 | 15.43 | 13.84 |
| TransCal | **4.84** | 7.44 | 6.84 | 6.38 | 4.07 | 8.14 | **3.04** | 2.81 | **4.67** | 8.23 | 5.13 | 5.65 | 4.76 | 5.18 | **3.74** | 7.04 | 7.23 |
| Ensemble | 10.92 | 4.98 | 3.29 | 6.40 | 13.30 | 18.89 | 3.81 | 5.75 | 9.48 | 17.31 | 14.56 | 6.25 | 5.49 | 8.77 | 14.82 | 8.23 | 14.02 |
| PseudoCal (ours) | 6.58 | **4.78** | **3.04** | **4.80** | **3.04** | **4.13** | 7.92 | 5.51 | 5.85 | **7.54** | **4.22** | 5.97 | 5.33 | **5.17** | 6.71 | **5.16** | **4.46** |
| Oracle | 3.21 | 3.26 | 2.17 | 2.88 | 1.00 | 2.21 | 2.90 | 1.75 | 2.29 | 1.82 | 2.11 | 3.55 | 1.76 | 2.47 | 0.99 | 2.47 | 1.06 |
| Accuracy (%) | 66.03 | 87.15 | 87.17 | 80.12 | 75.24 | 68.95 | 89.96 | 88.55 | 82.49 | 73.91 | 67.07 | 86.14 | 85.53 | 79.58 | 72.18 | 82.35 | 75.25 |

Table 4: ECE (%) of closed-set UDA on *DomainNet* (*DNet*).

| Method | ATDOC | | | | | BNM | | | | | MCC | | | | |
|---|---|---|---|---|---|---|---|---|---|---|---|---|---|---|---|
| | →C | →P | →R | →S | avg | →C | →P | →R | →S | avg | →C | →P | →R | →S | avg |
| No Calib. | 9.54 | 7.38 | 3.75 | 12.29 | 8.24 | 28.57 | 22.10 | 15.37 | 31.27 | 24.33 | 8.63 | 7.77 | 4.79 | 13.61 | 8.70 |
| TempScal-src | 8.69 | 7.71 | 1.94 | 11.82 | 7.54 | 19.04 | 13.62 | 9.40 | 20.30 | 15.59 | 8.38 | 8.32 | **2.36** | 13.88 | 8.23 |
| CPCS | 10.78 | 4.72 | 4.46 | 13.38 | 8.34 | 8.23 | 7.92 | 7.98 | 9.29 | 8.36 | 9.03 | 4.33 | 3.44 | 17.21 | 8.50 |
| TransCal | 23.02 | 24.76 | 26.65 | 19.68 | 23.52 | **6.52** | **1.84** | **5.82** | 9.39 | **5.89** | 22.27 | 24.06 | 23.45 | 18.03 | 21.95 |
| Ensemble | 6.32 | 4.54 | **1.59** | 9.05 | 5.37 | 23.44 | 18.61 | 12.61 | 26.21 | 20.22 | 5.71 | 5.10 | 2.57 | 10.34 | 5.93 |
| PseudoCal (ours) | **1.82** | **1.41** | 2.51 | **1.70** | **1.86** | 10.27 | 6.01 | 6.18 | **5.86** | 7.08 | **1.35** | **1.89** | 2.38 | **3.10** | **2.18** |
| Oracle | 1.55 | 0.94 | 0.86 | 1.07 | 1.10 | 2.40 | 1.66 | 3.40 | 1.30 | 2.19 | 1.16 | 1.44 | 1.09 | 0.89 | 1.14 |
| Accuracy (%) | 56.05 | 60.64 | 74.95 | 52.08 | 60.93 | 56.62 | 63.13 | 74.30 | 52.25 | 61.57 | 50.89 | 57.74 | 71.62 | 46.39 | 56.66 |

| Method | CDAN | | | | | SAFN | | | | | MCD | | | | | *DNet* |
|---|---|---|---|---|---|---|---|---|---|---|---|---|---|---|---|---|
| | →C | →P | →R | →S | avg | →C | →P | →R | →S | avg | →C | →P | →R | →S | avg | AVG |
| No Calib. | 10.17 | 9.64 | 5.56 | 14.44 | 9.95 | 17.94 | 14.44 | 10.15 | 21.26 | 15.95 | 9.56 | 7.40 | 3.80 | 12.93 | 8.42 | 12.60 |
| TempScal-src | 7.92 | 8.31 | 2.75 | 12.30 | 7.82 | 9.61 | 8.15 | 4.12 | 14.18 | 9.02 | 6.48 | 6.96 | 4.06 | 11.20 | 7.18 | 9.23 |
| CPCS | 10.75 | 4.28 | 5.57 | 6.91 | 6.88 | 10.92 | 5.91 | 8.22 | 22.59 | 11.91 | 7.02 | 3.51 | 1.96 | 21.79 | 8.57 | 8.76 |
| TransCal | 20.92 | 21.41 | 22.93 | 16.93 | 20.55 | 10.75 | 12.88 | 14.28 | 14.28 | 13.05 | 21.48 | 24.99 | 27.45 | 18.95 | 23.22 | 17.72 |
| Ensemble | 7.21 | 6.74 | 3.54 | 11.29 | 7.20 | 16.59 | 13.25 | 9.08 | 19.52 | 14.61 | 7.25 | 5.27 | 2.86 | 11.34 | 6.68 | 10.00 |
| PseudoCal (ours) | **1.58** | **1.89** | **1.86** | **2.67** | **2.00** | **3.33** | **1.30** | **1.50** | **2.76** | **2.22** | **2.27** | **1.16** | **1.01** | **1.70** | **1.53** | **2.81** |
| Oracle | 1.45 | 1.08 | 1.07 | 0.94 | 1.13 | 1.43 | 0.92 | 1.21 | 0.72 | 1.07 | 1.33 | 0.97 | 0.56 | 0.68 | 0.88 | 1.25 |
| Accuracy (%) | 53.11 | 59.13 | 71.82 | 49.09 | 58.29 | 49.59 | 58.03 | 66.40 | 47.66 | 55.42 | 48.85 | 57.99 | 65.32 | 47.95 | 55.03 | 57.98 |

Table 5: ECE (%) of partial-set UDA on *Office-Home* (*Home*).

| Method | ATDOC | | | | | MCC | | | | | PADA | | | | | *Home* |
|---|---|---|---|---|---|---|---|---|---|---|---|---|---|---|---|---|
| | →Ar | →Cl | →Pr | →Re | avg | →Ar | →Cl | →Pr | →Re | avg | →Ar | →Cl | →Pr | →Re | avg | AVG |
| No Calib. | 16.68 | 28.47 | 20.00 | 12.26 | 19.35 | 12.71 | 22.17 | 12.21 | 8.99 | 14.02 | 9.45 | 19.09 | 9.19 | 6.77 | 11.13 | 14.83 |
| TempScal-src | 13.40 | 24.79 | 14.91 | 8.72 | 15.45 | 7.12 | 15.97 | 6.04 | **4.35** | 8.37 | 8.92 | 18.20 | 6.21 | 4.08 | 9.35 | 11.06 |
| CPCS | 19.39 | 29.74 | 13.86 | 14.63 | 19.41 | 12.73 | 28.11 | 9.09 | 10.69 | 15.16 | 24.40 | 22.74 | 17.30 | 27.67 | 23.03 | 19.20 |
| TransCal | 10.64 | **5.17** | **5.88** | 11.30 | 8.25 | 9.44 | 4.27 | **5.41** | 6.98 | 6.53 | 22.70 | 11.00 | 23.00 | 26.77 | 20.87 | 11.88 |
| Ensemble | 11.98 | 21.28 | 13.44 | 8.62 | 13.83 | 9.22 | 18.54 | 10.11 | 6.88 | 10.46 | 5.30 | 11.86 | **4.43** | **3.92** | **6.38** | 10.46 |
| PseudoCal (ours) | **7.87** | 10.90 | 6.24 | **4.83** | **7.46** | **3.74** | **3.63** | 6.93 | 4.81 | **4.78** | **4.72** | **3.45** | 10.77 | 6.69 | 6.41 | **6.22** |
| Oracle | 4.13 | 4.45 | 4.37 | 4.08 | 4.26 | 2.81 | 3.01 | 3.06 | 2.37 | 2.81 | 3.94 | 2.65 | 4.80 | 3.03 | 3.61 | 3.56 |
| Accuracy (%) | 63.02 | 50.70 | 65.92 | 73.71 | 63.34 | 65.53 | 51.68 | 73.41 | 78.23 | 67.21 | 55.65 | 44.06 | 61.23 | 66.54 | 56.87 | 62.47 |

**Partial-set UDA.** We evaluate 3 partial-set UDA methods on *Office-Home* and report the ECE in Table 5. PseudoCal consistently performs the best on average and outperforms the second-best method (Ensemble) by a significant margin of $4.24\%$.

**Source-free UDA.** We evaluate source-free UDA settings using SHOT and DINE. We report the ECE for *DomainNet* and *Image-Sketch* together in Table 6. PseudoCal outperforms Ensemble on both benchmarks by significant margins, with $7.44\%$ on *DomainNet* and $15.05\%$ on *Image-Sketch*.

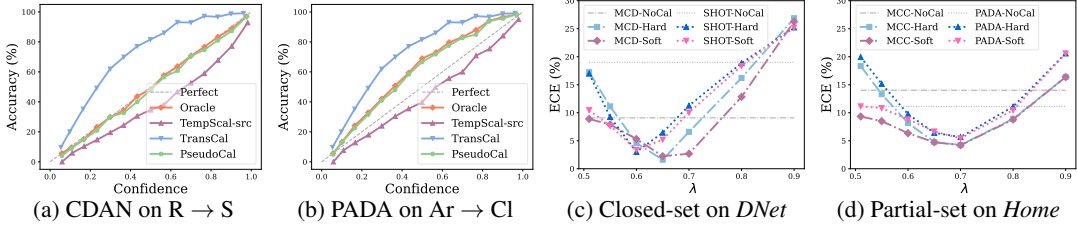

| (a) CDAN on R → S | (b) PADA on Ar → Cl | (c) Closed-set on *DNet* | (d) Partial-set on *Home* |

Figure 3: (a) and (b) provide the reliability diagrams of various calibration methods for a qualitative comparison. (c) and (d) present the sensitivity analysis of the fixed mix ratio $\lambda$.

Table 6: ECE (%) of source-free UDA on *DomainNet* (*DNet*) and *ImageNet-Sketch* (*Sketch*).

| Method | SHOT | | | | | | DINE | | | | | | *DNet* | *Sketch* |
| | →C | →P | →R | →S | avg | I→S | →C | →P | →R | →S | avg | I→S | AVG | AVG |
|---|---|---|---|---|---|---|---|---|---|---|---|---|---|---|
| No Calib. | 17.16 | 21.19 | 10.03 | 23.14 | 17.88 | 34.71 | 21.99 | 22.51 | 12.39 | 30.34 | 21.81 | 58.85 | 19.84 | 46.78 |
| Ensemble | 14.24 | 17.94 | 7.81 | 19.49 | 14.87 | 33.03 | 17.88 | 18.86 | 10.83 | 25.33 | 18.22 | 53.24 | 16.54 | 43.14 |
| PseudoCal (ours) | **6.66** | **7.78** | **2.91** | **6.67** | **6.00** | **8.42** | **14.42** | **12.95** | **5.30** | **16.15** | **12.20** | **47.76** | **9.10** | **28.09** |
| Oracle | 3.27 | 2.52 | 1.37 | 2.18 | 2.33 | 4.39 | 1.75 | 1.80 | 1.29 | 1.37 | 1.55 | 5.90 | 1.94 | 5.14 |
| Accuracy (%) | 66.52 | 64.48 | 78.34 | 59.64 | 67.25 | 34.29 | 63.76 | 65.47 | 80.69 | 55.51 | 66.36 | 22.27 | 66.80 | 28.28 |

**Semantic segmentation.** In addition to assessing the performance of PseudoCal in classification tasks, we also evaluate PseudoCal on the domain adaptive semantic segmentation tasks and report the ECE in Table 7. Remarkably, PseudoCal performs the best on average and demonstrates an average ECE improvement of $4.62\%$ over the no-calibration baseline.

Table 7: ECE (%) of segmentation.

| Method | *GTA5* | *SYNTHIA* | AVG |
|---|---|---|---|
| No Calib. | 7.87 | 23.08 | 15.48 |
| TempScal-src | 4.61 | 19.24 | 11.93 |
| Ensemble | **2.66** | 20.84 | 11.75 |
| PseudoCal (ours) | 5.73 | **15.99** | **10.86** |
| Oracle | 0.52 | 4.50 | 2.51 |

### 4.3 DISCUSSION

**Qualitative comparisons.** Reliability diagrams in Figure 3(a)-(b) show that PseudoCal consistently aligns with 'Oracle', while the existing state-of-the-art method TransCal deviates significantly.

**Impact of mix ratio $\lambda$.** The fixed mix ratio $\lambda$ is the sole hyperparameter in PseudoCal. We investigate its impact on calibration performance by experimenting with values ranging from $0.51$ to $0.9$. The results of two closed-set UDA methods (including SHOT) on *DomainNet* are presented in Figure 3(c), and the results of two partial-set UDA methods on *Office-Home* are shown in Figure 3(d). We first examine *mixup* with both 'Hard' (one-hot labels) and 'Soft' (soft predictions) labels, finding similar trends with differences that are generally not visible when $\lambda > 0.6$. In addition, optimal performance for PseudoCal occurs with a moderate $\lambda$ value between $0.6$ and $0.7$. The reason for this is that a $\lambda$ value closer to $0.5$ generates more ambiguous samples, resulting in increased wrong predictions, whereas a $\lambda$ value closer to $1.0$ has the opposite effect. For more details, kindly refer to Appendix D, where we examine the impact of different $\lambda$ values on the sample-level correspondence. At last, for simplicity, we use a fixed $\lambda$ value of $0.65$ with hard labels for all experiments.

**Impact of backbones and metrics.** In order to examine the robustness of PseudoCal across different backbones and calibration metrics, we assess its performance using ViT-B (Dosovitskiy et al., 2021) as the backbone and present the results for the aforementioned three metrics in Table 8. The findings reveal that PseudoCal consistently achieves the best performance regardless of the choice of backbone or calibration metric.

Table 8: ViT results of MCC on C→S.

| Method | ECE (%) | BS | NLL |
|---|---|---|---|
| No Calib. | 11.52 | 0.5674 | 1.9592 |
| TempScal-src | 10.63 | 0.5647 | 1.9418 |
| CPCS | 5.48 | 0.5579 | 1.8781 |
| TransCal | 23.38 | 0.6279 | 2.1089 |
| Ensemble | 10.08 | 0.5618 | 1.9260 |
| PseudoCal (ours) | **3.63** | **0.5553** | **1.8697** |
| Oracle | 1.29 | 0.5519 | 1.8597 |

**Impact of UDA model quality.** We've provided the target-domain accuracy for each UDA model in the 'Accuracy' row. PseudoCal remains effective as long as the UDA model has learned the target data structure instead of being completely randomly initialized, supported by the robust *cluster assumption*. This is evident in Table 6, where PseudoCal maintains its competence even with low accuracy pseudo-labels (about $30\%$).

Table 9: ECE (%) of ablation experiments on pseudo-target synthesis.

| Method | MCD | | BNM | | CDAN | SHOT | PADA | | DINE |
|---|---|---|---|---|---|---|---|---|---|
| | D→A | W→A | Cl→Pr | Pr→Re | R→C | I→S | Ar→Cl | Re→Ar | P→R |
| No Calib. | 16.39 | 17.03 | 22.09 | 15.72 | 9.83 | 34.71 | 20.35 | 8.31 | 12.39 |
| MocoV2Aug (Chen et al., 2020) | 16.85 | 17.21 | 20.51 | 14.98 | 15.49 | 28.63 | 25.81 | 15.17 | 11.12 |
| RandAug (Cubuk et al., 2020) | 12.87 | 11.53 | 19.24 | 11.37 | 13.33 | 29.28 | 18.47 | 10.32 | 12.62 |
| CutMix (Yun et al., 2019) | 8.20 | 6.39 | 14.82 | 10.60 | 7.60 | 23.18 | 15.96 | 6.04 | 6.93 |
| ManifoldMix (Verma et al., 2019) | 19.49 | 19.27 | 23.29 | 16.94 | 27.00 | 50.54 | 36.04 | 21.29 | 16.88 |
| Mixup-Beta (Zhang et al., 2018) | 14.96 | 13.11 | 15.65 | 11.24 | 15.84 | 26.74 | 23.85 | 11.46 | 9.72 |
| Pseudo-Label (Lee et al., 2013) | 32.47 | 33.35 | 26.31 | 19.65 | 47.02 | 65.70 | 56.18 | 36.27 | 19.31 |
| Filtered-PL (Sohn et al., 2020) | 31.74 | 32.73 | 26.14 | 19.46 | 45.35 | 64.29 | 54.83 | 35.10 | 19.05 |
| PseudoCal-same | 19.31 | 20.54 | 22.50 | 15.63 | 25.43 | 45.54 | 30.30 | 18.46 | 15.56 |
| PseudoCal (ours) | **4.38** | **4.06** | **6.31** | **4.76** | **1.51** | **8.42** | **2.95** | **3.71** | **5.29** |
| Oracle | 2.31 | 1.90 | 3.14 | 1.10 | 1.28 | 4.39 | 2.16 | 2.87 | 1.29 |
| Accuracy (%) | 67.52 | 66.63 | 73.69 | 80.35 | 52.98 | 34.29 | 43.82 | 63.73 | 80.69 |

**Comparison with training-stage *mixup*.** Most approaches incorporate *mixup* (Zhang et al., 2018) during the model training stage as an objective to enhance model generalization, and among them, Thulasidasan et al. (2019) further utilizes *mixup* as a training-stage calibration method. However, our use of *mixup* in PseudoCal differs significantly from previous *mixup*-based works in three key aspects. *(i)* Different stages: All of these works apply *mixup* in training, while our *mixup* operation occurs in the inference stage to synthesize a labeled set. *(ii)* Different mix ratios: PseudoCal leverages *mixup* for cross-cluster sample interpolation and performs effectively with a fixed mix ratio $\lambda \in (0.6, 0.7)$ but is considerably less effective with $\lambda$ values close to 1.0. In contrast, previous methods typically work best with $\lambda \in \text{Beta}(\alpha, \alpha)$ where $\alpha \in [0.1, 0.4]$, essentially favoring $\lambda$ values that are close to 1.0. However, they are ineffective with $\lambda$ values close to 0.5 (like our adopted values) due to the manifold intrusion problem (Thulasidasan et al., 2019; Guo et al., 2019). *(iii)* Different performance: We observed that UDA models trained with training-time calibration methods still suffer from significant miscalibration, while our PseudoCal can further substantially reduce ECE errors for these models. For example, as shown in Table 6, SHOT employs *label smoothing*(Müller et al., 2019; Liu et al., 2022) during training, and DINE is trained with *mixup*(Thulasidasan et al., 2019; Verma et al., 2022).

**Ablation study on pseudo-target synthesis.** Pseudo-target synthesis plays a critical role in our PseudoCal framework. In this step, we employ input-level *mixup* with a fixed mix ratio ($\lambda$) to generate a pseudo-target sample by combining two real samples with different pseudo-labels. We conduct a comprehensive ablation study on this synthesis strategy by extensively comparing it with alternative approaches, including: *(i)* Applying *mixup* between samples with the same pseudo-label (referred to as PseudoCal-same). *(ii)* Using instance-based augmentations of target samples, such as RandAugment (Cubuk et al., 2020), and strong augmentations commonly used in self-supervised learning (Chen et al., 2020). *(iii)* Employing *mixup* at different levels, such as the patch-level (Yun et al., 2019) and the feature-level (Verma et al., 2019). *(iv)* Applying common training-stage *mixup* using $\lambda \in \text{Beta}(0.3, 0.3)$ (Zhang et al., 2018). *(v)* Directly utilizing original or filtered pseudo-labeled real target samples (Lee et al., 2013; Sohn et al., 2020) without *mixup* (by setting the mix ratio $\lambda$ to 1.0). We present an extensive comparison of all these strategies in Table 9. The results consistently demonstrate that our inference-stage input-level *mixup* outperforms the alternative options.

## 5 CONCLUSION

In conclusion, we introduce PseudoCal, a novel and versatile post-hoc framework for calibrating predictive uncertainty in unsupervised domain adaptation (UDA). By focusing on the unlabeled target domain, PseudoCal distinguishes itself from mainstream calibration methods that are based on *covariate shift* and eliminates their associated limitations. To elaborate, PseudoCal employs a novel inference-stage *mixup* strategy to synthesize a labeled pseudo-target set that mimics the correct-wrong statistics in real target samples. In this way, PseudoCal successfully transforms the challenging unsupervised calibration problem involving unlabeled real samples into a supervised one using labeled pseudo-target data, which can be readily addressed through *temperature scaling*. Throughout our extensive evaluations spanning diverse UDA settings beyond *covariate shift*, including source-free UDA settings and domain adaptive semantic segmentation, PseudoCal consistently showcases its advantages of simplicity, versatility, and effectiveness in enhancing calibration in UDA. In future work, we aim to extend PseudoCal to address calibration problems in more practical UDA scenarios, including open-set UDA and UDA for object detection.

## REPRODUCIBILITY STATEMENT

Our PseudoCal approach is simple and does not require any model training or hyperparameter tuning (with a fixed hyperparameter). We have provided detailed information in the implementation section. Importantly, we have included a comprehensive PyTorch-style pseudocode in Appendix A, covering every algorithm detail and step necessary for implementing our approach. Furthermore, we plan to release a full implementation upon acceptance.

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

# A  ALGORITHM

The PyTorch-style pseudocode for our calibration method PseudoCal is provided in Algorithm 1.

---

**Algorithm 1** PyTorch-style pseudocode for PseudoCal.

---

```python
# x: A batch of real target images with shuffled order.
# lam: The mix ratio, a fixed scalar value between 0.5 and 1.0.
# net: A trained UDA model in the evaluation mode.

# Perform pseudo-target synthesis for a mini-batch.
def pseudo_target_synthesis(x, lam, net):

    # Use the random index within the data batch
    # to obtain a pair of real samples for mixup.
    rand_idx = torch.randperm(x.shape[0])
    inputs_a = x
    inputs_b = x[rand_idx]

    # Obtain model predictions and pseudo labels (pl).
    pred_a = net(inputs_a)
    pl_a = pred_a.max(dim=1)[1]
    pl_b = pl_a[rand_idx]

    # Select the samples with distinct labels for the mixup.
    diff_idx = (pl_a != pl_b).nonzero(as_tuple=True)[0]

    # Mixup with images and labels.
    pseudo_target_x = lam * inputs_a + (1 - lam) * inputs_b

    # If the user is not aware that lam is between 0.5 and 1.0,
    # the following if-else code can avoid bugs.
    if lam > 0.5:
        pseudo_target_y = pl_a
    else:
        pseudo_target_y = pl_b

    return pseudo_target_x[diff_idx], pseudo_target_y[diff_idx]

# Perform supervised calibration using pseudo-target data.
def pseudoCal(x, lam, net):

    # Synthesize a mini-batch of pseudo-target samples and labels.
    pseudo_x, pseudo_y = pseudo_target_synthesis(x, lam, net)

    # Infer the logits for the pseudo-target samples.
    pseudo_logit = net(pseudo_x)

    # Apply temperature scaling to estimate the
    # pseudo-target temperature as the real temperature.
    calib_method = TempScaling()
    pseudo_temp = calib_method(pseudo_logit, pseudo_y)

    return pseudo_temp
```

---

# B  ADDITIONAL CALIBRATION METRICS

In addition to the Expected Calibration Error (ECE) (Guo et al., 2017) discussed in the main text, we also consider two other calibration metrics as follows. Let $\mathbf{y}_i$ represent the one-hot ground truth encoding for input sample $x_i$, and $\hat{\mathbf{p}}_i$ denote the predicted probability vector output by the model $\phi$.

**Negative Log-Likelihood (NLL)** (Goodfellow et al., 2016) is also known as the cross-entropy loss. The NLL loss for a single sample $x_i$ is given by:

$$\mathcal{L}_{\text{NLL}} = -\sum_{c=1}^{C} \mathbf{y}_i^c \log \hat{\mathbf{p}}_i^c$$

**Brier Score (BS)** (Brier et al., 1950) can be defined as the squared error between the predicted probability vector and the one-hot label vector. The Brier Score for a single sample $x_i$ is given by:

$$\mathcal{L}_{\text{BS}} = \frac{1}{C} \sum_{c=1}^{C} (\hat{\mathbf{p}}_i^c - \mathbf{y}_i^c)^2$$

In addition to the ViT results presented in the main text, we have observed consistent advantages of our PseudoCal method over existing calibration methods across all three calibration metrics: ECE, NLL, and BS. We choose to report the ECE results for most of the experiments as ECE (Guo et al., 2017) is a widely used calibration metric.

## C  SEMANTIC SEGMENTATION CALIBRATION DETAILS

For our calibration experiments on semantic segmentation, we calibrate the models trained solely on the source domain (GTA5 (Richter et al., 2016) or SYNTHIA (Ros et al., 2016)) without any target adaptation. We treat each pixel as an individual sample in classification tasks for both *mixup* and *temperature scaling*. To address the computational complexity, we adopt the evaluation strategy suggested in previous studies (de Jorge et al., 2023) and randomly sample 20,000 pixels from each image (with resolutions such as 1920*720) for calibration.

## D  ANALYSIS OF SAMPLE-LEVEL CORRESPONDENCE IN PSEUDOCAL

In the **Analysis** part of Section 3.2 in the main text, we offer an intuitive analysis of the sample-level correspondence between pseudo-target data and real target samples. Figure 1(b) qualitatively illustrates the striking similarity in the correct-wrong statistics between the real target and pseudo target. To further enhance the understanding of this correspondence, we aim for a quantitative sample-level analysis. Consider a pair of real samples $x_t^i$ with pseudo-label $\hat{y}_t^i$ inferred by the UDA model $\phi$, and $x_t^j$ with pseudo-label $\hat{y}_t^j$. We employ the *mixup* operation to generate a mixed sample $x_{\text{pt}}^i$ with the mixed label $y_{\text{pt}}^i$ using Equation 3. For simplicity, we assume that all labels are one-hot hard labels and $\lambda$ is in the range of $(0.5, 1.0)$. This implies that $x_t^i$ functions as the primary real sample, directly determining the mixed label $y_{\text{pt}}^i$, i.e., $y_{\text{pt}}^i == \hat{y}_t^i$. We apply the *mixup* operation $n_t$ times during the model inference stage using unlabeled target data. This results in a labeled pseudo-target set $\{(x_{\text{pt}}^i, y_{\text{pt}}^i)\}_{i=1}^{n_t}$ and the original pseudo-labeled real target set $\{(x_t^i, \hat{y}_t^i)\}_{i=1}^{n_t}$. Using the same UDA model $\phi$, we infer predictions $\hat{y}_{\text{pt}}^i$ for the mixed sample $x_{\text{pt}}^i$ and traverse through all mixed samples. For the mixed pseudo-target samples, we obtain predictions $\{\hat{y}_{\text{pt}}^i\}_{i=1}^{n_t}$ and corresponding labels $\{y_{\text{pt}}^i\}_{i=1}^{n_t}$. Regarding real target samples, the predictions are the available pseudo-labels $\{\hat{y}_t^i\}_{i=1}^{n_t}$, while the labels are ground truth labels $\{y_t^i\}_{i=1}^{n_t}$ which are used to assess the UDA model accuracy.

$$\text{CR}_{\text{correct}} = \frac{\sum_i^{n_t} (\hat{y}_{\text{pt}}^i == y_{\text{pt}}^i) \cdot (\hat{y}_t^i == y_t^i)}{\sum_i^{n_t} (\hat{y}_t^i == y_t^i)} \tag{4}$$

$$\text{CR}_{\text{wrong}} = \frac{\sum_i^{n_t} (\hat{y}_{\text{pt}}^i \neq y_{\text{pt}}^i) \cdot (\hat{y}_t^i \neq y_t^i)}{\sum_i^{n_t} (\hat{y}_t^i \neq y_t^i)} \tag{5}$$

$$\text{CR}_{\text{arithmetic}} = \frac{\sum_i^{n_t} (\hat{y}_{\text{pt}}^i == y_{\text{pt}}^i) \cdot (\hat{y}_t^i == y_t^i) + \sum_i^{n_t} (\hat{y}_{\text{pt}}^i \neq y_{\text{pt}}^i) \cdot (\hat{y}_t^i \neq y_t^i)}{n_t} \tag{6}$$

$$\text{CR}_{\text{harmonic}} = \frac{2 \cdot \text{CR}_{\text{correct}} \cdot \text{CR}_{\text{wrong}}}{\text{CR}_{\text{correct}} + \text{CR}_{\text{wrong}}} \tag{7}$$

Using these predictions and labels, we can systematically quantify the sample-level correspondence between the pseudo and real target sets for a more in-depth understanding. We establish such

correspondence when both the predictions of a mixed pseudo sample and its primary real sample are either both correct or both wrong, as assessed by their respective labels. In other words, we consider a correspondence when $\hat{y}_{\text{pt}}^i == y_{\text{pt}}^i$ and $\hat{y}_{\text{t}}^i == y_{\text{t}}^i$, or when $\hat{y}_{\text{pt}}^i \neq y_{\text{pt}}^i$ and $\hat{y}_{\text{t}}^i \neq y_{\text{t}}^i$. To quantitatively measure this sample-level correspondence, we introduce four correspondence metrics. The first metric, denoted as $\text{CR}_{\text{correct}}$, represents the correspondence rate of correct real samples (see Equation 4). It indicates how many correct real samples maintain correspondence with their mixed counterparts. Similarly, our second metric, denoted as $\text{CR}_{\text{wrong}}$, measures the correspondence rate of wrong real samples (see Equation 5). For a more comprehensive perspective, we introduce the third metric, $\text{CR}_{\text{arithmetic}}$, which calculates the arithmetic mean of $\text{CR}_{\text{correct}}$ and $\text{CR}_{\text{wrong}}$, assessing the correspondence rate of all real samples (see Equation 6). However, it's important to note that these three metrics may be misleading in extreme situations where most of the correspondences are biased toward either being correct or wrong. To address this issue, we propose our fourth metric, $\text{CR}_{\text{harmonic}}$, which takes the harmonic mean of $\text{CR}_{\text{correct}}$ and $\text{CR}_{\text{wrong}}$, providing equal consideration to both correct and wrong correspondences (see Equation 7). This metric is inspired by the success of the H-Score solution in balanced accuracy measurement for known-unknown accuracy in open-set UDA, as demonstrated by previous studies (Fu et al., 2020; Bucci et al., 2020).

Table 10: By tuning the mix ratio $\lambda$, we can synthesize the most ambiguous pseudo samples ($\lambda = 0.51$) and the simplest ones ($\lambda = 1.0$), i.e., the pseudo-labeled real samples themselves. PseudoCal employs a moderate value of $\lambda = 0.65$ for all the results. Under these three cases, we measure the sample-level correspondence between the real samples and pseudo samples using four correspondence metrics.

| Method | MCD | | BNM | | CDAN | SHOT | PADA | | DINE |
|---|---|---|---|---|---|---|---|---|---|
| | D→A | W→A | Cl→Pr | Pr→Re | R→C | I→S | Ar→Cl | Re→Ar | P→R |
| No Calib. ECE (%) | 16.39 | 17.03 | 22.09 | 15.72 | 9.83 | 34.71 | 20.35 | 8.31 | 12.39 |
| PseudoCal ($\lambda$=1.0) | 32.47 | 33.35 | 26.31 | 19.65 | 47.02 | 65.70 | 56.18 | 36.27 | 19.31 |
| PseudoCal ($\lambda$=0.65) | **4.38** | **4.06** | 6.31 | **4.76** | **1.51** | 8.42 | 2.95 | 3.71 | **5.29** |
| PseudoCal ($\lambda$=0.51) | 13.77 | 11.69 | 11.85 | 14.13 | 15.15 | 11.08 | 11.03 | 23.07 | 14.50 |
| Oracle ECE (%) | 2.31 | 1.90 | 3.14 | 1.10 | 1.28 | 4.39 | 2.16 | 2.87 | 1.29 |
| Accuracy (%) | 67.52 | 66.63 | 73.69 | 80.35 | 52.98 | 34.29 | 43.82 | 63.73 | 80.69 |
| # of correct real data | 1826 | 1792 | 3183 | 3408 | 9650 | 17218 | 703 | 656 | 55757 |
| # of wrong real data | 872 | 894 | 1135 | 836 | 8548 | 32998 | 918 | 385 | 13342 |
| $\text{CR}_{\text{harmonic}}$ ($\lambda$=1.0) (%) | 0 | 0 | 0 | 0 | 0 | 0 | 0 | 0 | 0 |
| $\text{CR}_{\text{harmonic}}$ ($\lambda$=0.65) (%) | **63.45** | **63.45** | **59.89** | **59.27** | **60.56** | **56.28** | **60.21** | **62.04** | **61.73** |
| $\text{CR}_{\text{harmonic}}$ ($\lambda$=0.51) (%) | 52.08 | 54.42 | 53.13 | 52.87 | 45.33 | 35.18 | 50.94 | 46.03 | 56.26 |
| $\text{CR}_{\text{arithmetic}}$ ($\lambda$=1.0) (%) | **67.68** | **66.72** | **73.71** | **80.30** | 53.03 | 34.29 | 43.37 | **63.02** | **80.69** |
| $\text{CR}_{\text{arithmetic}}$ ($\lambda$=0.65) (%) | 62.36 | 62.75 | 61.72 | 63.08 | **61.58** | 65.58 | **63.92** | 61.00 | 70.73 |
| $\text{CR}_{\text{arithmetic}}$ ($\lambda$=0.51) (%) | 52.07 | 54.10 | 50.03 | 47.51 | 56.35 | **66.48** | 63.02 | 50.52 | 50.74 |
| $\text{CR}_{\text{correct}}$ ($\lambda$=1.0) (%) | **100** | **100** | **100** | **100** | **100** | **100** | **100** | **100** | **100** |
| $\text{CR}_{\text{correct}}$ ($\lambda$=0.65) (%) | 59.93 | 61.16 | 63.52 | 65.22 | 52.09 | 44.48 | 50.74 | 56.02 | 75.11 |
| $\text{CR}_{\text{correct}}$ ($\lambda$=0.51) (%) | 38.53 | 41.35 | 41.69 | 40.54 | 30.90 | 21.88 | 36.67 | 31.94 | 44.76 |
| $\text{CR}_{\text{wrong}}$ ($\lambda$=1.0) (%) | 0 | 0 | 0 | 0 | 0 | 0 | 0 | 0 | 0 |
| $\text{CR}_{\text{wrong}}$ ($\lambda$=0.65) (%) | 67.40 | 65.92 | 56.66 | 54.32 | 72.31 | 76.60 | 74.04 | 69.52 | 52.40 |
| $\text{CR}_{\text{wrong}}$ ($\lambda$=0.51) (%) | **80.32** | **79.57** | **73.2** | **75.99** | **85.04** | **89.75** | **83.38** | **82.39** | **75.73** |

For empirical illustration, we conduct experiments using PseudoCal with varied $\lambda$ values of $\{0.51, 0.65, 1.0\}$, among which $0.65$ is our default value for all experiments in the main text. We report all results, including the measurement results of the sample-level correspondence using the four metrics described above, in Table 10. From the shown results, we can make three consistent observations: *(i)* As expected, only the harmonic metric $\text{CR}_{\text{harmonic}}$ is reliable and aligns with the actual calibration performance, while both the one-sided correct measure $\text{CR}_{\text{correct}}$ and the wrong measure $\text{CR}_{\text{wrong}}$ can be extremely biased, which would further directly mislead the arithmetic mean metric $\text{CR}_{\text{arithmetic}}$. *(ii)* In line with the discussion on the impact of mix ratio ($\lambda$) in Section 4.3, our observations reveal that $\lambda$ values near $0.5$ predominantly yield wrong predictions for pseudo-target samples (mixed samples), while $\lambda$ values of $1.0$ result in entirely correct predictions. The role of $\lambda$ in controlling cross-cluster perturbation, determining the difficulty of mixed samples, is noteworthy. A $\lambda$ close to $0.5$ generates ambiguous mixed samples with almost even contributions from two real samples bearing different pseudo-labels. In such instances, the UDA model struggles to ascertain the class label, resulting in predominantly wrong predictions when evaluated with mixed labels. Conversely, a $\lambda$ of $1.0$ equates to not using *mixup* and directly leveraging pseudo-labeled real target samples. This scenario constitutes the easiest mixed samples, as the UDA model outputs predictions

identical to raw target predictions, leading to entirely correct predictions when assessed with target pseudo-labels. From the *cluster assumption* perspective, extreme $\lambda$ values render the relevant analysis inconclusive. A $\lambda$ value very close to $0.5$ makes it challenging to determine the primary real sample. Conversely, a $\lambda$ value very close to $1.0$ signifies the negligible cross-cluster perturbation, generating a mixed sample nearly identical to the primary real sample, wherein the cluster assumption does not apply. In general, extreme $\lambda$ values, whether close to $0.5$ or $1.0$, exhibit significant bias towards either wrong or correct predictions, which indicates correct-wrong statistics of the pseudo-target set become skewed, deviating from real target samples. Hence, for a typical UDA model with both correct and wrong target predictions, we recommend employing a moderate $\lambda$ value, such as the $0.65$ utilized in our main text. *(iii)* Taking a closer look at the reliable measure of sample-level correspondence by $\mathrm{CR}_{\mathrm{harmonic}}$, we find that for various UDA models, there maintains a high correspondence with a $\mathrm{CR}_{\mathrm{harmonic}}$ value of about $60\%$, even for a low-accuracy model with only $30\%$ accuracy. This strongly supports the robust existence of the *cluster assumption* and the robustness of our analysis in Section 3.2. For a vivid illustration of the impact of $\lambda$ values on sample-level correspondence, Figure 4 presents the correct-wrong statistics of all UDA methods outlined in Table 10. We find that extreme $\lambda$ values result in a notable skewness in the correct-wrong statistics of the pseudo-target set when compared to the real target set. For a clear visualization of mixed images, please see Figure 5.

## E    LIMITATIONS AND BROADER IMPACTS

PseudoCal has the following limitations and potential negative societal impacts: *(i)* Like other calibration methods compared, PseudoCal may occasionally increase ECE when the initial ECE is already small (see →D in Table 3), which raises risks for safety-critical decision-making systems. *(ii)* PseudoCal may face challenges in extreme cases with very few available unlabeled target samples, such as only a small batch of samples or even a single target sample. *(iii)* PseudoCal is partly dependent on the *cluster assumption*, and it may fail if the target pseudo label is extremely poor, i.e., performing similarly to random trials.

Table 11: ECE (%) of calibration results when combining PseudoCal with different supervised calibration methods, including MatrixScal (Guo et al., 2017), VectorScal (Guo et al., 2017), and TempScal (Guo et al., 2017) (our default choice).

| Method | MCD | | BNM | | CDAN | SHOT | PADA | | DINE |
|---|---|---|---|---|---|---|---|---|---|
| | D→A | W→A | Cl→Pr | Pr→Re | R→C | I→S | Ar→Cl | Re→Ar | P→R |
| No Calib. | 16.39 | 17.03 | 22.09 | 15.72 | 9.83 | 34.71 | 20.35 | 8.31 | 12.39 |
| MatrixScal-src | 17.86 | 20.28 | 25.73 | 15.98 | 22.11 | - | 36.55 | 20.45 | - |
| VectorScal-src | 17.75 | 20.52 | 16.40 | 12.36 | 12.88 | - | 20.53 | 9.07 | - |
| TempScal-src | 32.09 | 18.65 | 15.10 | 11.64 | 9.27 | - | 15.15 | 6.34 | - |
| PseudoCal(Matrix.) | 11.61 | 13.20 | 16.07 | 11.83 | 15.09 | 42.86 | 35.85 | 27.07 | 7.65 |
| PseudoCal(Vector.) | 11.00 | 9.32 | 9.31 | 6.05 | 6.37 | 23.90 | 5.90 | 4.19 | 6.23 |
| PseudoCal(Temp.) | **4.38** | **4.06** | **6.31** | **4.76** | **1.51** | **8.42** | **2.95** | **3.71** | **5.29** |
| Oracle | 2.31 | 1.90 | 3.14 | 1.10 | 1.28 | 4.39 | 2.16 | 2.87 | 1.29 |
| Accuracy (%) | 67.52 | 66.63 | 73.69 | 80.35 | 52.98 | 34.29 | 43.82 | 63.73 | 80.69 |

## F    FULL CALIBRATION RESULTS

Due to space constraints in the main text, we have presented the average results for tasks with the same target domain. For example, in the case of *Office-Home*, UDA tasks including 'Cl→Ar', 'Pr→Ar', and 'Re→Ar' share the common target domain 'Ar'. Consequently, we have averaged the results of these three UDA tasks and reported the averaged value in the tables within our main text under the row labeled '→ Ar'. Additionally, note that the 'avg' row represents the averaged results within each UDA method's rows to the left of the 'avg' row. Differently, the 'AVG' row signifies the averaged results across all 'avg' rows associated with different UDA methods. Consequently, the 'AVG' row can be considered more reliable and representative for drawing conclusions.

Additionally, as *matrix scaling* (MatrixScal), *vector scaling* (VectorScal), and *temperature scaling* (TempScal) are similar, all proposed by Guo et al. (2017), and the authors have demonstrated that *temperature scaling* (TempScal) is the superior solution. Therefore, as for the source-domain

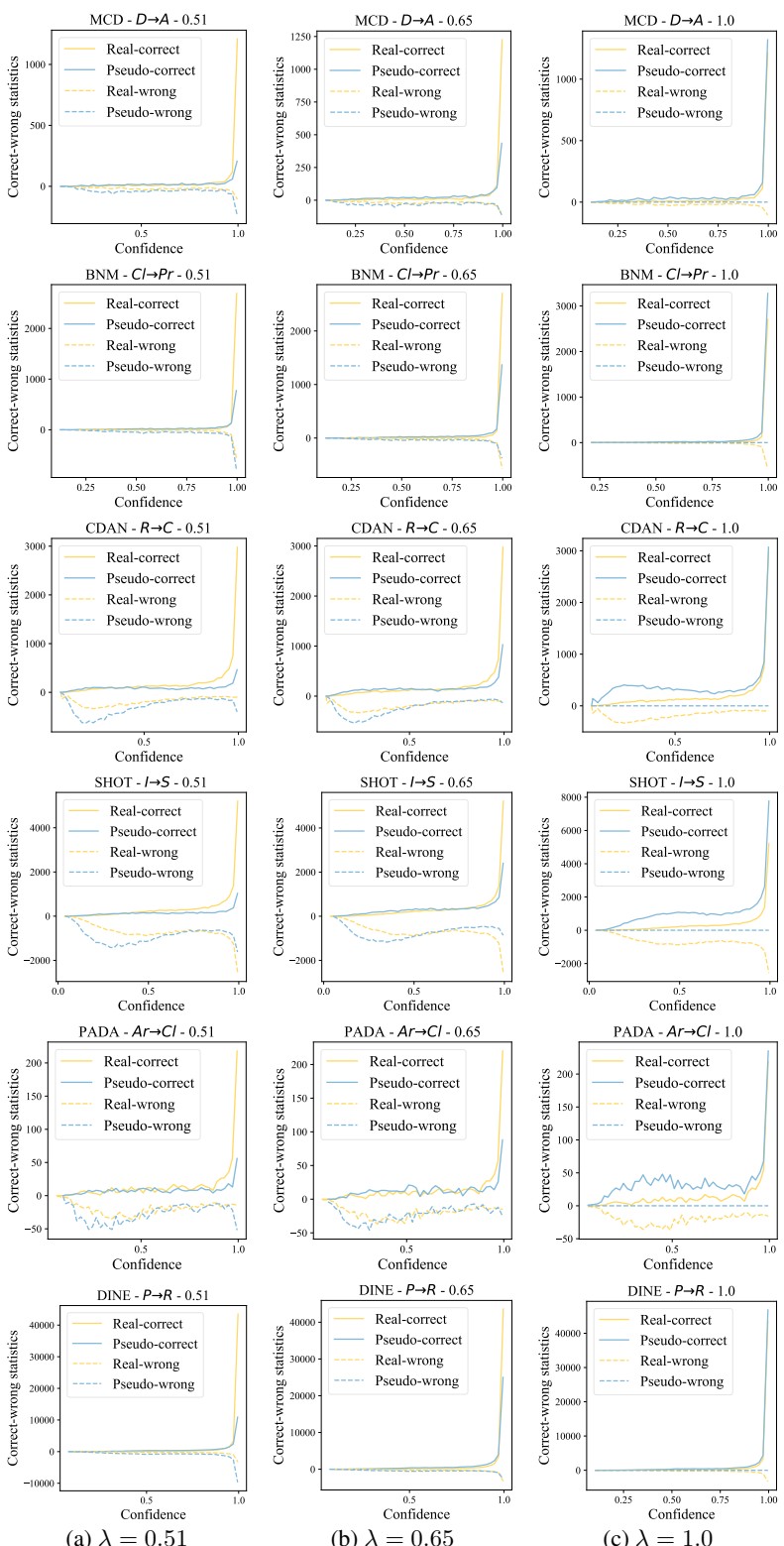

(a) $\lambda = 0.51$      (b) $\lambda = 0.65$      (c) $\lambda = 1.0$

Figure 4: The correct-wrong statistics are computed for both the pseudo-target and real target sets. We partition confidence values into 50 bins and present the count of correct and wrong predictions in each bin. Correctness for real target data is determined by comparing predictions of real target samples with ground truths. For pseudo-target data, correctness is assessed by comparing predictions of the mixed samples with mixed labels.

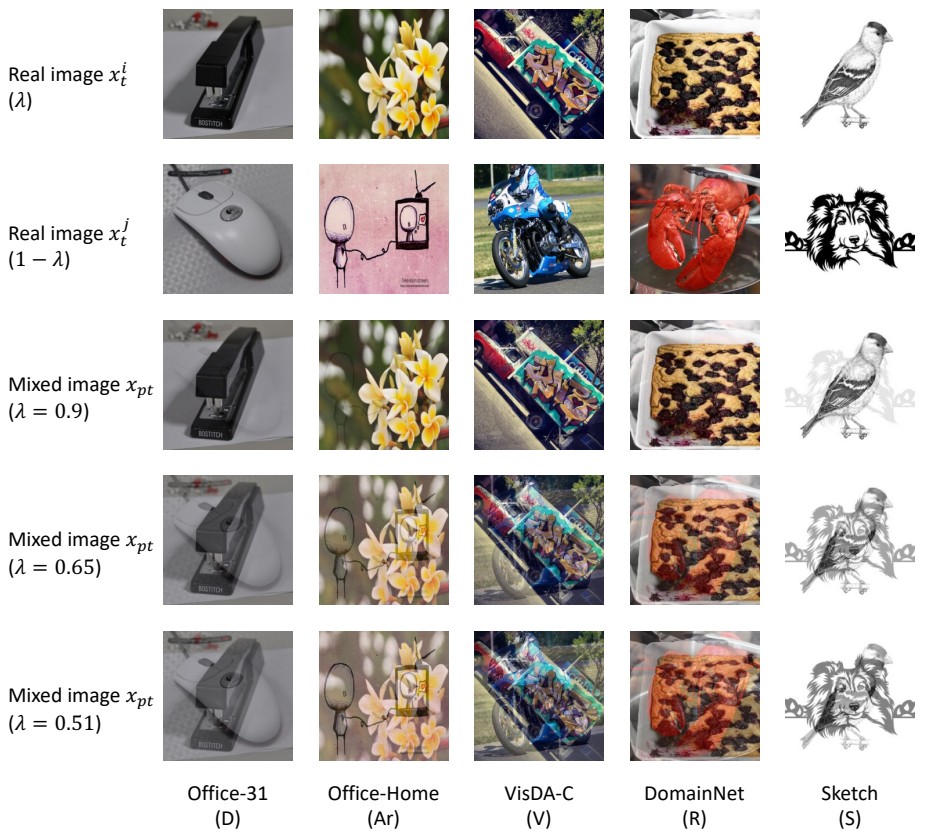

| | Office-31 (D) | Office-Home (Ar) | VisDA-C (V) | DomainNet (R) | Sketch (S) |
|---|---|---|---|---|---|

Figure 5: Visualization of input-level *mixup* for various UDA benchmarks with varied $\lambda$ values.

calibration baseline (using a labeled source validation set for calibration), we have only reported the results of TempScal-src in the tables in the main text. Here, we present the results of MatrixScal-src and VectorScal-src for additional reference, without impacting any of the conclusions drawn in the main text. While our PseudoCal is inspired by the factorized NLL of TempScal and naturally employs TempScal as the default supervised calibration method for our synthesized labeled pseudo-target set, we investigate the compatibility of PseudoCal with alternative supervised calibration methods, such as MatrixScal and VectorScal. The corresponding results are detailed in Table 11. Our findings reveal two key observations: *(i)* If a supervised calibration method exhibits stability and effectiveness with the source labeled data, combining it with PseudoCal tends to yield reduced ECE error compared to the no calibration baseline. *(ii)* Due to the similarity in correct-wrong statistics between the pseudo-target set and real target data, PseudoCal demonstrates compatibility with both MatrixScal and VectorScal. However, it consistently achieves the best calibration performance when paired with TempScal, aligning with the conclusion in (Guo et al., 2017) that TempScal generally outperforms MatrixScal and VectorScal. For detailed calibration results for each task, please refer to Table 12 through Table 30.

Table 12: ECE (%) of a closed-set UDA method ATDOC (Liang et al., 2021) on *Office-Home*.

| Method | Ar → Cl | Ar → Pr | Ar → Re | Cl → Ar | Cl → Pr | Cl → Re | Pr → Ar | Pr → Cl | Pr → Re | Re → Ar | Re → Cl | Re → Pr | avg |
|---|---|---|---|---|---|---|---|---|---|---|---|---|---|
| No Calib. | 22.83 | 10.57 | 6.31 | 10.77 | 8.88 | 6.38 | 10.39 | 22.61 | 5.49 | 9.06 | 21.61 | 6.38 | 11.77 |
| MatrixScal-src | 35.03 | 20.72 | 18.28 | 27.54 | 24.73 | 23.40 | 22.51 | 32.85 | 13.66 | 20.25 | 32.89 | 12.90 | 23.73 |
| VectorScal-src | 22.05 | 10.09 | 5.85 | 11.51 | 7.74 | 6.01 | 15.12 | 26.85 | 7.81 | 7.94 | 21.10 | 5.03 | 12.26 |
| TempScal-src | 14.69 | 5.55 | **2.60** | 4.27 | **3.17** | 1.45 | 9.67 | 22.55 | 5.04 | 4.63 | 15.37 | **3.21** | 7.68 |
| CPCS | 8.37 | 9.32 | 6.44 | 12.94 | 14.94 | 11.41 | 12.28 | 6.00 | 4.13 | 17.18 | 29.88 | 8.80 | 11.81 |
| TransCal | 4.95 | 13.85 | 16.58 | 17.29 | 17.34 | 18.76 | 18.77 | 7.48 | 19.54 | 18.20 | 7.13 | 16.90 | 14.73 |
| Ensemble | 18.40 | 7.47 | 4.51 | 7.82 | 4.76 | 4.24 | 8.36 | 17.96 | **3.92** | 5.96 | 17.68 | 4.29 | 8.78 |
| PseudoCal | **3.07** | **4.23** | 5.28 | **1.96** | 6.27 | 5.70 | **2.52** | **4.05** | 4.22 | **2.79** | **1.68** | 7.03 | **4.07** |
| Oracle | 2.38 | 3.14 | 2.34 | 1.44 | 1.92 | 1.36 | 1.98 | 1.92 | 1.37 | 1.71 | 1.43 | 1.80 | 1.90 |
| Accuracy (%) | 52.07 | 74.48 | 79.27 | 64.24 | 73.85 | 75.42 | 64.65 | 50.65 | 78.54 | 70.37 | 54.46 | 81.48 | 68.29 |

Table 13: ECE (%) of a closed-set UDA method BNM (Cui et al., 2020) on *Office-Home*.

| Method | Ar → Cl | Ar → Pr | Ar → Re | Cl → Ar | Cl → Pr | Cl → Re | Pr → Ar | Pr → Cl | Pr → Re | Re → Ar | Re → Cl | Re → Pr | avg |
|---|---|---|---|---|---|---|---|---|---|---|---|---|---|
| No Calib. | 38.64 | 22.49 | 16.21 | 30.89 | 22.09 | 18.25 | 34.90 | 42.46 | 15.72 | 27.11 | 38.44 | 14.52 | 26.81 |
| MatrixScal-src | 39.37 | 23.31 | 19.01 | 30.30 | 25.73 | 22.24 | 31.37 | 41.37 | 15.98 | 24.06 | 37.39 | 14.77 | 27.07 |
| VectorScal-src | 30.83 | 17.66 | 9.97 | 21.91 | 16.40 | 11.46 | 27.76 | 37.27 | 12.36 | 18.91 | 29.06 | 10.03 | 20.30 |
| TempScal-src | 27.22 | 16.34 | 8.91 | 20.39 | 15.10 | 10.21 | 28.82 | 35.60 | 11.64 | 20.12 | 28.15 | 9.67 | 19.35 |
| CPCS | 33.80 | 18.08 | 8.12 | 17.24 | 19.77 | 7.90 | 28.68 | **17.28** | 10.39 | 28.36 | 23.97 | 6.86 | 18.37 |
| TransCal | 25.75 | 12.11 | 5.87 | 15.73 | 10.51 | 5.51 | 21.41 | 29.66 | 5.02 | **15.17** | 26.25 | 4.80 | 14.82 |
| Ensemble | 29.52 | 16.03 | 12.00 | 22.77 | 15.55 | 14.06 | 25.17 | 32.06 | 11.53 | 19.55 | 30.46 | 11.56 | 20.02 |
| PseudoCal | **14.27** | **8.74** | **4.60** | **15.46** | **6.31** | **4.69** | **20.90** | 18.35 | **4.76** | 15.66 | **15.47** | **3.55** | **11.06** |
| Oracle | 3.16 | 2.18 | 1.76 | 2.00 | 3.14 | 1.95 | 2.92 | 1.78 | 1.10 | 1.68 | 2.64 | 1.77 | 2.17 |
| Accuracy (%) | 54.39 | 73.49 | 79.78 | 64.52 | 73.69 | 76.82 | 61.68 | 51.13 | 80.35 | 70.05 | 55.56 | 82.36 | 68.65 |

Table 14: ECE (%) of a closed-set UDA method MCC (Jin et al., 2020) on *Office-Home*.

| Method | Ar → Cl | Ar → Pr | Ar → Re | Cl → Ar | Cl → Pr | Cl → Re | Pr → Ar | Pr → Cl | Pr → Re | Re → Ar | Re → Cl | Re → Pr | avg |
|---|---|---|---|---|---|---|---|---|---|---|---|---|---|
| No Calib. | 23.74 | 14.31 | 10.89 | 12.70 | 13.15 | 11.72 | 14.36 | 23.18 | 8.98 | 12.69 | 22.40 | 9.54 | 14.81 |
| MatrixScal-src | 37.39 | 23.28 | 19.95 | 31.00 | 27.75 | 25.27 | 26.13 | 35.70 | 16.27 | 21.56 | 35.20 | 14.95 | 26.20 |
| VectorScal-src | 21.05 | 12.79 | 7.87 | 10.96 | 11.18 | 8.20 | 16.87 | 28.29 | 9.64 | 7.58 | 21.40 | 6.15 | 13.50 |
| TempScal-src | 12.23 | 6.43 | 3.61 | 4.06 | 4.69 | 2.85 | 11.38 | 22.91 | 5.83 | 4.79 | 13.60 | **4.11** | 8.04 |
| CPCS | 25.11 | 15.31 | **3.60** | 19.41 | 14.36 | 4.49 | 13.83 | 35.66 | 8.56 | 24.08 | 24.99 | 14.27 | 16.97 |
| TransCal | 3.04 | 6.31 | 5.98 | 12.75 | 7.42 | 8.60 | 11.95 | 4.59 | 9.90 | 10.48 | 3.95 | 6.37 | 7.61 |
| Ensemble | 19.20 | 11.30 | 8.05 | 10.01 | 9.69 | 8.51 | 10.11 | 18.98 | 7.13 | 9.15 | 19.42 | 7.44 | 11.58 |
| PseudoCal | **2.71** | **5.04** | 3.81 | **3.17** | **4.64** | 3.06 | **2.66** | **1.54** | 3.85 | **2.73** | **2.51** | 5.86 | **3.47** |
| Oracle | 2.41 | 2.57 | 2.31 | 2.67 | 1.73 | 1.62 | 1.58 | 0.84 | 1.80 | 2.51 | 1.66 | 2.35 | 2.00 |
| Accuracy (%) | 47.26 | 69.29 | 75.90 | 59.91 | 68.33 | 70.16 | 56.32 | 44.49 | 76.04 | 66.87 | 50.65 | 79.48 | 63.73 |

Table 15: ECE (%) of a closed-set UDA method CDAN (Long et al., 2018) on *Office-Home*.

| Method | Ar → Cl | Ar → Pr | Ar → Re | Cl → Ar | Cl → Pr | Cl → Re | Pr → Ar | Pr → Cl | Pr → Re | Re → Ar | Re → Cl | Re → Pr | avg |
|---|---|---|---|---|---|---|---|---|---|---|---|---|---|
| No Calib. | 24.88 | 14.66 | 10.39 | 14.71 | 13.05 | 11.25 | 13.24 | 22.54 | 8.37 | 12.19 | 21.41 | 8.74 | 14.62 |
| MatrixScal-src | 35.03 | 22.64 | 19.14 | 28.14 | 26.14 | 22.64 | 24.20 | 33.34 | 15.03 | 20.32 | 30.69 | 13.78 | 24.28 |
| VectorScal-src | 18.81 | 10.46 | 7.24 | 8.92 | 9.81 | 6.73 | 15.31 | 26.51 | 9.18 | 7.51 | 16.70 | 5.76 | 11.91 |
| TempScal-src | 12.48 | 5.82 | 3.40 | **5.57** | 5.14 | 3.06 | 9.78 | 21.29 | 6.12 | 5.31 | 12.55 | **4.06** | 7.88 |
| CPCS | 31.45 | 13.21 | 2.36 | 25.84 | 24.68 | 17.24 | 13.44 | 27.86 | 10.09 | 15.85 | 41.38 | 7.98 | 19.28 |
| TransCal | **2.65** | 11.04 | 11.67 | 14.44 | 13.41 | 14.01 | 16.34 | 6.04 | 15.50 | 13.51 | 5.46 | 11.77 | 11.32 |
| Ensemble | 18.64 | 11.85 | 7.23 | 10.87 | 9.04 | 7.94 | 9.45 | 19.12 | 6.52 | 9.90 | 17.97 | 6.56 | 11.26 |
| PseudoCal | 3.52 | **4.33** | **2.32** | 5.67 | **4.81** | 2.82 | **6.36** | **3.78** | **2.05** | **3.28** | **3.85** | 5.00 | **3.98** |
| Oracle | 1.83 | 2.96 | 1.94 | 3.88 | 1.74 | 2.20 | 4.46 | 3.22 | 1.68 | 2.50 | 3.48 | 2.08 | 2.66 |
| Accuracy (%) | 48.00 | 67.00 | 75.07 | 59.83 | 66.88 | 69.98 | 58.59 | 48.64 | 76.31 | 68.36 | 53.33 | 79.68 | 64.31 |

Table 16: ECE (%) of a closed-set UDA method SAFN (Xu et al., 2019) on *Office-Home*.

| Method | Ar → Cl | Ar → Pr | Ar → Re | Cl → Ar | Cl → Pr | Cl → Re | Pr → Ar | Pr → Cl | Pr → Re | Re → Ar | Re → Cl | Re → Pr | avg |
|---|---|---|---|---|---|---|---|---|---|---|---|---|---|
| No Calib. | 28.25 | 15.29 | 12.40 | 16.62 | 14.10 | 12.45 | 18.17 | 29.68 | 10.94 | 14.92 | 25.77 | 10.08 | 17.39 |
| MatrixScal-src | 37.63 | 23.66 | 20.05 | 28.07 | 26.01 | 23.00 | 25.60 | 37.84 | 16.22 | 20.98 | 33.18 | 14.69 | 25.58 |
| VectorScal-src | 21.01 | 12.78 | 9.20 | 10.96 | 10.28 | 7.67 | 16.03 | 26.93 | 8.91 | 10.72 | 20.21 | 6.35 | 13.42 |
| TempScal-src | 12.33 | 5.56 | 3.17 | 4.62 | 4.22 | **3.40** | 9.99 | 21.72 | 5.64 | 6.36 | 14.33 | 3.89 | 7.94 |
| CPCS | 31.45 | 16.18 | 10.90 | 23.93 | 11.19 | 6.71 | 15.78 | 25.66 | 18.73 | 5.24 | 34.50 | **2.80** | 16.92 |
| TransCal | 7.50 | **4.23** | **2.80** | 4.11 | **3.63** | 4.89 | 3.14 | 7.47 | 4.76 | 3.26 | 5.65 | 3.46 | 4.57 |
| Ensemble | 25.00 | 13.33 | 9.91 | 15.20 | 11.62 | 10.14 | 16.12 | 26.14 | 9.54 | 13.15 | 23.56 | 8.55 | 15.19 |
| PseudoCal | **3.30** | 6.41 | 4.14 | **3.46** | 7.06 | 5.18 | **2.99** | **3.40** | 3.79 | **2.70** | **3.33** | 7.12 | **4.41** |
| Oracle | 3.10 | 3.78 | 1.94 | 2.06 | 1.85 | 2.18 | 2.65 | 1.66 | 1.11 | 1.16 | 2.68 | 1.92 | 2.17 |
| Accuracy (%) | 50.65 | 70.96 | 75.81 | 64.44 | 70.42 | 72.30 | 62.55 | 49.55 | 77.16 | 70.54 | 55.51 | 79.97 | 66.66 |

Table 17: ECE (%) of a closed-set UDA method MCD (Saito et al., 2018) on *Office-Home*.

| Method | Ar → Cl | Ar → Pr | Ar → Re | Cl → Ar | Cl → Pr | Cl → Re | Pr → Ar | Pr → Cl | Pr → Re | Re → Ar | Re → Cl | Re → Pr | avg |
|---|---|---|---|---|---|---|---|---|---|---|---|---|---|
| No Calib. | 26.24 | 16.26 | 12.30 | 16.42 | 14.19 | 13.27 | 19.02 | 27.38 | 10.35 | 13.63 | 24.25 | 9.43 | 16.89 |
| MatrixScal-src | 41.44 | 28.57 | 22.89 | 34.21 | 27.91 | 26.19 | 28.46 | 39.91 | 18.20 | 22.91 | 36.82 | 16.58 | 28.67 |
| VectorScal-src | 21.79 | 12.62 | 8.36 | 11.89 | 7.19 | 7.75 | 17.75 | 27.43 | 8.99 | 10.10 | 20.83 | 5.72 | 13.37 |
| TempScal-src | 8.59 | **4.59** | **2.87** | 3.65 | **2.79** | **2.90** | 10.42 | 17.99 | 4.85 | 3.96 | 9.86 | **3.29** | 6.31 |
| CPCS | 20.66 | 11.43 | 21.72 | 27.95 | 11.22 | 11.03 | 24.03 | 12.63 | 10.13 | 23.42 | 48.48 | 7.86 | 19.21 |
| TransCal | **2.43** | 8.94 | 9.45 | 10.78 | 10.81 | 10.80 | 9.86 | **2.07** | 13.56 | 11.69 | 3.49 | 11.19 | 8.76 |
| Ensemble | 20.49 | 10.59 | 7.24 | 11.59 | 9.53 | 9.16 | 15.53 | 22.66 | 6.52 | 9.95 | 19.45 | 6.66 | 12.45 |
| PseudoCal | 2.52 | 4.93 | 3.93 | **3.39** | 6.57 | 3.70 | **5.05** | 2.68 | **3.52** | **3.76** | **3.39** | 7.28 | **4.23** |
| Oracle | 2.22 | 2.48 | 2.08 | 2.68 | 2.31 | 2.13 | 3.02 | 1.97 | 2.44 | 2.26 | 2.61 | 2.11 | 2.36 |
| Accuracy (%) | 46.55 | 63.75 | 73.01 | 57.44 | 64.86 | 67.45 | 53.81 | 42.77 | 73.72 | 65.88 | 51.07 | 77.63 | 61.49 |

Table 18: ECE (%) of a closed-set UDA method ATDOC (Liang et al., 2021) on *DomainNet*.

| Method | C → S | P → C | P → R | R → C | R → P | R → S | S → P | avg |
|---|---|---|---|---|---|---|---|---|
| No Calib. | 12.22 | 9.27 | 3.75 | 9.81 | 6.85 | 12.36 | 7.92 | 8.88 |
| MatrixScal-src | 34.30 | 27.58 | 15.58 | 23.23 | 18.37 | 28.05 | 27.44 | 24.94 |
| VectorScal-src | 16.19 | 11.45 | 3.97 | 15.11 | 10.19 | 19.26 | 9.52 | 12.24 |
| TempScal-src | 10.32 | 6.52 | 1.94 | 10.86 | 8.51 | 13.31 | 6.92 | 8.34 |
| CPCS | 12.87 | 13.31 | 4.46 | 8.25 | 5.11 | 13.90 | 4.34 | 8.89 |
| TransCal | 19.89 | 23.51 | 26.65 | 22.52 | 24.93 | 19.46 | 24.59 | 23.08 |
| Ensemble | 8.71 | 5.73 | **1.59** | 6.91 | 4.41 | 9.38 | 4.66 | 5.91 |
| PseudoCal | **1.68** | **1.98** | 2.51 | **1.66** | **1.21** | **1.71** | **1.61** | **1.77** |
| Oracle | 0.98 | 1.92 | 0.86 | 1.18 | 0.70 | 1.16 | 1.17 | 1.14 |
| Accuracy (%) | 53.74 | 56.51 | 74.95 | 55.59 | 61.65 | 50.41 | 59.64 | 58.93 |

Table 19: ECE (%) of a closed-set UDA method BNM (Cui et al., 2020) on *DomainNet*.

| Method | C → S | P → C | P → R | R → C | R → P | R → S | S → P | avg |
|---|---|---|---|---|---|---|---|---|
| No Calib. | 30.88 | 29.27 | 15.37 | 27.87 | 21.79 | 31.65 | 22.41 | 25.61 |
| MatrixScal-src | 37.91 | 31.17 | 18.31 | 26.82 | 22.33 | 32.31 | 28.64 | 28.21 |
| VectorScal-src | 23.10 | 20.02 | 9.88 | 21.80 | 14.83 | 26.68 | 14.18 | 18.64 |
| TempScal-src | 19.11 | 18.79 | 9.40 | 19.28 | 14.42 | 21.49 | 12.81 | 16.47 |
| CPCS | 14.45 | 13.75 | 7.98 | **2.72** | 4.35 | **4.14** | 11.50 | 8.41 |
| TransCal | 9.21 | **6.31** | **5.82** | 6.73 | **1.69** | 9.56 | **1.98** | **5.90** |
| Ensemble | 25.08 | 23.46 | 12.61 | 23.42 | 18.52 | 27.34 | 18.70 | 21.30 |
| PseudoCal | **5.08** | 12.43 | 6.18 | 8.10 | 5.20 | 6.64 | 6.82 | 7.21 |
| Oracle | 1.60 | 3.17 | 3.40 | 1.63 | 1.50 | 1.00 | 1.81 | 2.02 |
| Accuracy (%) | 52.90 | 55.52 | 74.30 | 57.71 | 63.95 | 51.61 | 62.30 | 59.76 |

Table 20: ECE (%) of a closed-set UDA method MCC (Jin et al., 2020) on *DomainNet*.

| Method | C → S | P → C | P → R | R → C | R → P | R → S | S → P | avg |
|---|---|---|---|---|---|---|---|---|
| No Calib. | 15.19 | 8.29 | 4.79 | 8.98 | 6.91 | 12.04 | 8.63 | 9.26 |
| MatrixScal-src | 36.95 | 28.60 | 15.99 | 23.92 | 18.95 | 29.54 | 28.72 | 26.10 |
| VectorScal-src | 18.52 | 11.63 | 4.49 | 15.98 | 10.72 | 20.86 | 10.71 | 13.27 |
| TempScal-src | 13.49 | 5.92 | **2.36** | 10.83 | 8.96 | 14.27 | 7.67 | 9.07 |
| CPCS | 29.26 | 15.02 | 3.44 | 3.03 | 6.00 | 5.15 | 2.66 | 9.22 |
| TransCal | 16.89 | 22.54 | 23.45 | 22.00 | 24.68 | 19.17 | 23.44 | 21.74 |
| Ensemble | 11.36 | 5.38 | 2.57 | 6.03 | 4.40 | 9.32 | 5.80 | 6.41 |
| PseudoCal | **2.72** | **1.45** | 2.38 | **1.25** | **1.64** | **3.48** | **2.13** | **2.15** |
| Oracle | 0.80 | 1.36 | 1.09 | 0.96 | 1.18 | 0.97 | 1.70 | 1.15 |
| Accuracy (%) | 47.65 | 51.27 | 71.62 | 50.51 | 59.02 | 45.14 | 56.46 | 54.52 |

Table 21: ECE (%) of a closed-set UDA method CDAN (Long et al., 2018) on *DomainNet*.

| Method | C → S | P → C | P → R | R → C | R → P | R → S | S → P | avg |
|---|---|---|---|---|---|---|---|---|
| No Calib. | 17.00 | 10.51 | 5.56 | 9.83 | 8.26 | 11.88 | 11.03 | 10.58 |
| MatrixScal-src | 35.28 | 27.82 | 15.80 | 22.11 | 18.34 | 27.24 | 27.76 | 24.91 |
| VectorScal-src | 17.44 | 10.88 | 4.37 | 12.88 | 9.45 | 17.90 | 9.81 | 11.82 |
| TempScal-src | 13.39 | 6.58 | 2.75 | 9.27 | 8.30 | 11.22 | 8.32 | 8.55 |
| CPCS | **2.40** | 17.27 | 5.57 | 4.24 | 6.75 | 11.42 | **1.81** | 7.07 |
| TransCal | 14.85 | 20.65 | 22.93 | 21.19 | 22.27 | 19.01 | 20.55 | 20.21 |
| Ensemble | 12.96 | 7.47 | 3.54 | 6.96 | 5.73 | 9.62 | 7.75 | 7.72 |
| PseudoCal | 3.48 | **1.65** | **1.86** | **1.51** | **1.70** | **1.85** | 2.08 | **2.02** |
| Oracle | 1.03 | 1.61 | 1.07 | 1.28 | 0.73 | 0.84 | 1.43 | 1.14 |
| Accuracy (%) | 49.07 | 53.25 | 71.82 | 52.98 | 60.75 | 49.11 | 57.51 | 56.36 |

Table 22: ECE (%) of a closed-set UDA method SAFN (Xu et al., 2019) on *DomainNet*.

| Method | C → S | P → C | P → R | R → C | R → P | R → S | S → P | avg |
|---|---|---|---|---|---|---|---|---|
| No Calib. | 21.82 | 17.98 | 10.15 | 17.90 | 13.63 | 20.70 | 15.25 | 16.78 |
| MatrixScal-src | 33.45 | 22.54 | 11.16 | 21.05 | 15.53 | 26.33 | 21.85 | 21.70 |
| VectorScal-src | 19.61 | 14.11 | 4.73 | 17.45 | 10.40 | 21.04 | 10.49 | 13.98 |
| TempScal-src | 15.12 | 8.37 | 4.12 | 10.86 | 8.23 | 13.25 | 8.07 | 9.72 |
| CPCS | 21.96 | 14.58 | 8.22 | 7.26 | 7.52 | 23.23 | 4.31 | 12.44 |
| TransCal | 6.58 | 11.28 | 14.28 | 10.21 | 12.67 | 7.18 | 13.10 | 10.76 |
| Ensemble | 19.74 | 16.66 | 9.08 | 16.51 | 12.48 | 19.31 | 14.03 | 15.40 |
| PseudoCal | **3.40** | **4.44** | **1.50** | **2.23** | **0.81** | **2.12** | **1.79** | **2.33** |
| Oracle | 0.86 | 1.75 | 1.21 | 1.11 | 0.78 | 0.57 | 1.06 | 1.05 |
| Accuracy (%) | 48.14 | 48.65 | 66.40 | 50.54 | 59.89 | 47.18 | 56.17 | 53.85 |

Table 23: ECE (%) of a closed-set UDA method MCD (Saito et al., 2018) on *DomainNet*.

| Method | C → S | P → C | P → R | R → C | R → P | R → S | S → P | avg |
|---|---|---|---|---|---|---|---|---|
| No Calib. | 12.97 | 9.47 | 3.80 | 9.65 | 7.01 | 12.89 | 7.80 | 9.08 |
| MatrixScal-src | 31.47 | 19.56 | 10.05 | 20.32 | 14.30 | 24.98 | 18.45 | 19.88 |
| VectorScal-src | 19.63 | 12.59 | 5.75 | 16.53 | 10.21 | 20.95 | 10.27 | 13.70 |
| TempScal-src | 11.61 | 5.39 | 4.06 | 7.58 | 7.19 | 10.79 | 6.74 | 7.62 |
| CPCS | 19.75 | 6.09 | 1.96 | 7.94 | 3.92 | 23.82 | 3.10 | 9.51 |
| TransCal | 19.44 | 21.53 | 27.45 | 21.44 | 25.19 | 18.45 | 24.79 | 22.61 |
| Ensemble | 11.60 | 7.54 | 2.86 | 6.95 | 5.35 | 11.07 | 5.19 | 7.22 |
| PseudoCal | **1.66** | **3.60** | **1.01** | **0.93** | **1.11** | **1.73** | **1.21** | **1.61** |
| Oracle | 0.62 | 1.81 | 0.56 | 0.85 | 0.91 | 0.73 | 1.03 | 0.93 |
| Accuracy (%) | 49.09 | 48.21 | 65.32 | 49.49 | 59.58 | 46.81 | 56.40 | 53.56 |

Table 24: ECE (%) of closed-set UDA methods on *Office-31*.

| Method | ATDOC (Liang et al., 2021) | | | | | BNM (Cui et al., 2020) | | | | | MCC (Jin et al., 2020) | | | | |
|---|---|---|---|---|---|---|---|---|---|---|---|---|---|---|---|
| | A → D | A → W | D → A | W → A | avg | A → D | A → W | D → A | W → A | avg | A → D | A → W | D → A | W → A | avg |
| No Calib. | 4.59 | 6.66 | 11.43 | 12.91 | 8.90 | 11.12 | 8.27 | 24.60 | 22.22 | 16.55 | 6.18 | 7.80 | 18.60 | 19.97 | 13.14 |
| MatrixScal-src | 9.58 | 13.21 | 14.04 | 15.35 | 13.05 | 11.22 | 8.81 | 24.64 | 21.94 | 16.65 | 9.70 | 10.21 | 18.99 | 21.84 | 15.19 |
| VectorScal-src | 4.57 | 6.43 | 15.69 | 17.50 | 11.05 | 8.15 | 4.11 | 24.82 | 23.59 | 15.17 | 5.12 | 3.16 | 20.53 | 24.01 | 13.21 |
| TempScal-src | **3.39** | 4.18 | 24.37 | 20.41 | 13.09 | 9.23 | 4.98 | 26.15 | 21.55 | 15.48 | 3.79 | 3.00 | 22.07 | 20.70 | 12.39 |
| CPCS | 7.98 | 8.94 | 26.49 | 22.80 | 16.55 | 11.65 | **2.02** | 27.16 | 17.73 | 14.64 | 4.69 | 3.03 | 29.84 | 30.47 | 17.01 |
| TransCal | 14.21 | 14.64 | 13.27 | 11.02 | 13.29 | **5.22** | 2.70 | 16.00 | 12.72 | 9.41 | 3.77 | 3.91 | 5.57 | 7.49 | 5.19 |
| Ensemble | 3.60 | **4.09** | 9.04 | 10.53 | 6.82 | 6.92 | 4.63 | 19.99 | 19.56 | 12.78 | 3.07 | 4.88 | 17.18 | 17.78 | 10.73 |
| PseudoCal | 6.64 | 4.98 | **3.22** | **4.47** | **4.83** | 6.30 | 3.97 | **10.75** | **8.21** | **7.31** | **2.68** | **2.82** | **4.50** | **4.71** | **3.68** |
| Oracle | 2.49 | 3.15 | 1.90 | 2.35 | 2.47 | 2.65 | 1.40 | 2.63 | 2.41 | 2.27 | 2.36 | 2.67 | 2.42 | 2.05 | 2.38 |
| Accuracy (%) | 91.57 | 88.93 | 73.41 | 73.06 | 81.74 | 88.35 | 90.94 | 71.35 | 73.77 | 81.10 | 91.37 | 89.06 | 69.86 | 69.51 | 79.95 |

Table 25: ECE (%) of closed-set UDA methods on *Office-31*.

| Method | CDAN (Long et al., 2018) | | | | | SAFN (Xu et al., 2019) | | | | | MCD (Saito et al., 2018) | | | | |
|---|---|---|---|---|---|---|---|---|---|---|---|---|---|---|---|
| | A → D | A → W | D → A | W → A | avg | A → D | A → W | D → A | W → A | avg | A → D | A → W | D → A | W → A | avg |
| No Calib. | 9.34 | 7.96 | 16.66 | 17.39 | 12.84 | 6.17 | 6.68 | 20.34 | 22.33 | 13.88 | 9.49 | 8.88 | 16.39 | 17.03 | 12.95 |
| MatrixScal-src | 11.90 | 14.91 | 17.21 | 21.12 | 16.29 | 9.49 | 13.97 | 20.56 | 23.43 | 16.86 | 9.83 | 13.49 | 17.86 | 20.28 | 15.37 |
| VectorScal-src | 6.04 | 3.60 | 17.67 | 25.37 | 13.17 | 3.22 | **2.20** | 21.07 | 23.59 | 12.52 | 5.87 | 4.61 | 17.75 | 20.52 | 12.19 |
| TempScal-src | 5.70 | 3.41 | 16.10 | 20.97 | 11.55 | 3.21 | 2.83 | 24.48 | 23.41 | 13.48 | **3.44** | **2.36** | 32.09 | 18.65 | 14.14 |
| CPCS | 30.95 | 5.67 | **4.99** | 29.95 | 17.89 | 8.21 | 18.21 | 24.18 | 22.12 | 18.18 | 11.85 | 19.01 | 32.45 | 22.92 | 21.56 |
| TransCal | 7.44 | 6.84 | 5.51 | **4.18** | 5.99 | **3.04** | 2.81 | 6.43 | 9.86 | 5.54 | 5.65 | 4.76 | 5.86 | 4.39 | 5.17 |
| Ensemble | 4.98 | 3.29 | 7.41 | 14.43 | 7.53 | 3.81 | 5.75 | 17.58 | 20.20 | 11.84 | 6.25 | 5.49 | 13.53 | 15.60 | 10.22 |
| PseudoCal | **4.78** | **3.04** | 6.39 | 6.78 | **5.25** | 7.92 | 5.51 | **4.00** | **4.26** | **5.42** | 5.97 | 5.33 | **4.38** | **4.06** | **4.94** |
| Oracle | 3.26 | 2.17 | 2.94 | 3.47 | 2.96 | 2.90 | 1.75 | 2.14 | 2.27 | 2.27 | 3.55 | 1.76 | 2.31 | 1.90 | 2.38 |
| Accuracy (%) | 87.15 | 87.17 | 64.82 | 67.23 | 76.59 | 89.96 | 88.55 | 69.33 | 68.58 | 79.11 | 86.14 | 85.53 | 67.52 | 66.63 | 76.46 |

Table 26: ECE (%) of a partial-set UDA method ATDOC (Liang et al., 2021) on *Office-Home*.

| Method | Ar → Cl | Ar → Pr | Ar → Re | Cl → Ar | Cl → Pr | Cl → Re | Pr → Ar | Pr → Cl | Pr → Re | Re → Ar | Re → Cl | Re → Pr | avg |
|---|---|---|---|---|---|---|---|---|---|---|---|---|---|
| No Calib. | 28.21 | 20.87 | 10.76 | 17.58 | 23.49 | 11.69 | 19.16 | 28.98 | 14.34 | 13.29 | 28.22 | 15.64 | 19.35 |
| MatrixScal-src | 35.85 | 19.37 | 13.42 | 29.69 | 30.20 | 21.94 | 21.96 | 37.00 | 14.83 | 19.36 | 34.96 | 16.94 | 24.63 |
| VectorScal-src | 25.87 | 15.83 | 7.46 | 18.37 | 20.96 | 11.63 | 19.96 | 33.03 | 12.36 | 11.16 | 26.57 | 11.61 | 17.90 |
| TempScal-src | 21.08 | 15.04 | 5.75 | 12.95 | 17.86 | 7.52 | 18.23 | 29.63 | 12.88 | 9.02 | 23.66 | 11.83 | 15.45 |
| CPCS | 28.34 | 27.40 | 19.28 | 14.37 | 6.27 | 10.86 | 32.51 | 39.04 | 13.75 | 11.28 | 21.84 | 7.92 | 19.41 |
| TransCal | **4.36** | **5.07** | 10.58 | 9.47 | **4.98** | 12.82 | **9.12** | **5.81** | 10.51 | 13.32 | **5.34** | 7.60 | 8.25 |
| Ensemble | 20.32 | 12.06 | 8.90 | 11.80 | 17.57 | 7.89 | 12.32 | 22.25 | 9.07 | 11.81 | 21.26 | 10.68 | 13.83 |
| PseudoCal | 9.15 | 7.08 | **3.21** | **7.59** | 7.53 | **4.84** | 11.80 | 12.79 | **6.45** | **4.21** | 10.75 | **4.10** | **7.46** |
| Oracle | 3.09 | 4.24 | 2.82 | 4.78 | 4.93 | 4.48 | 4.04 | 5.03 | 4.94 | 3.58 | 5.24 | 3.95 | 4.26 |
| Accuracy (%) | 51.46 | 64.99 | 77.19 | 61.89 | 61.34 | 73.44 | 59.50 | 49.01 | 70.51 | 67.68 | 51.64 | 71.43 | 63.34 |

Table 27: ECE (%) of a partial-set UDA method MCC (Jin et al., 2020) on *Office-Home*.

| Method | Ar → Cl | Ar → Pr | Ar → Re | Cl → Ar | Cl → Pr | Cl → Re | Pr → Ar | Pr → Cl | Pr → Re | Re → Ar | Re → Cl | Re → Pr | avg |
|---|---|---|---|---|---|---|---|---|---|---|---|---|---|
| No Calib. | 22.91 | 11.67 | 8.45 | 14.42 | 14.34 | 10.29 | 12.63 | 21.14 | 8.22 | 11.09 | 22.46 | 10.63 | 14.02 |
| MatrixScal-src | 35.16 | 19.13 | 14.89 | 29.94 | 30.26 | 25.30 | 24.67 | 34.81 | 14.78 | 18.58 | 34.09 | 15.73 | 24.78 |
| VectorScal-src | 19.52 | 9.73 | 6.05 | 12.79 | 14.23 | 11.07 | 16.13 | 26.53 | 9.03 | 9.29 | 20.18 | 7.95 | 13.54 |
| TempScal-src | 13.14 | **5.37** | **3.05** | 5.96 | 6.62 | **4.21** | 10.00 | 20.08 | 5.79 | 5.39 | 14.70 | 6.12 | 8.37 |
| CPCS | 19.34 | 10.62 | 4.00 | 4.25 | **4.14** | 12.00 | 28.24 | 37.75 | 16.08 | 5.70 | 27.24 | 12.51 | 15.16 |
| TransCal | 2.74 | 6.19 | 5.25 | 8.09 | 5.92 | 8.40 | 11.03 | 6.01 | 7.29 | 9.20 | **4.06** | **4.13** | 6.53 |
| Ensemble | 18.27 | 9.86 | 6.49 | 9.68 | 11.37 | 7.27 | 8.76 | 18.05 | 6.57 | 9.21 | 19.31 | 9.10 | 11.16 |
| PseudoCal | **2.51** | 7.86 | 4.70 | **3.04** | 6.70 | 5.78 | **4.20** | **4.01** | **3.96** | **3.99** | 4.36 | 6.23 | **4.78** |
| Oracle | 2.29 | 3.75 | 2.04 | 2.67 | 3.07 | 3.11 | 2.69 | 3.26 | 1.97 | 3.06 | 3.47 | 2.35 | 2.81 |
| Accuracy (%) | 51.10 | 74.17 | 81.56 | 62.53 | 66.72 | 73.16 | 63.27 | 50.03 | 79.96 | 70.80 | 53.91 | 79.33 | 67.21 |

Table 28: ECE (%) of a partial-set UDA method PADA (Cao et al., 2018) on *Office-Home*.

| Method | Ar → Cl | Ar → Pr | Ar → Re | Cl → Ar | Cl → Pr | Cl → Re | Pr → Ar | Pr → Cl | Pr → Re | Re → Ar | Re → Cl | Re → Pr | avg |
|---|---|---|---|---|---|---|---|---|---|---|---|---|---|
| No Calib. | 20.35 | 8.33 | 5.30 | 11.10 | 12.28 | 10.19 | 8.93 | 18.60 | 4.83 | 8.31 | 18.33 | 6.95 | 11.13 |
| MatrixScal-src | 36.55 | 24.04 | 16.23 | 34.97 | 33.22 | 28.87 | 27.26 | 37.58 | 16.54 | 20.45 | 35.41 | 16.45 | 27.30 |
| VectorScal-src | 20.53 | 7.22 | 4.71 | 12.28 | 13.91 | 13.44 | 22.41 | 31.95 | 9.35 | 9.07 | 19.86 | 8.57 | 14.44 |
| TempScal-src | 15.15 | 6.09 | 3.34 | 6.51 | 6.43 | **4.64** | 13.91 | 23.77 | 4.27 | 6.34 | 15.69 | 6.11 | 9.35 |
| CPCS | 24.22 | 30.26 | 24.81 | 9.80 | 7.37 | 43.23 | 28.84 | 39.45 | 14.97 | 34.57 | 4.55 | 14.27 | 23.03 |
| TransCal | 9.39 | 23.43 | 26.71 | 21.37 | 20.51 | 21.88 | 22.49 | 11.25 | 31.71 | 24.23 | 12.37 | 25.06 | 20.87 |
| Ensemble | 11.42 | **4.97** | **2.88** | 6.02 | **4.54** | 4.65 | **3.76** | 11.15 | **4.24** | 6.13 | 13.00 | **3.79** | **6.38** |
| PseudoCal | **2.95** | 12.31 | 7.51 | **4.68** | 10.14 | 5.38 | 5.77 | **4.13** | 7.19 | **3.71** | **3.28** | 9.85 | 6.41 |
| Oracle | 2.16 | 5.65 | 2.27 | 3.89 | 5.70 | 2.83 | 5.06 | 2.73 | 3.98 | 2.87 | 3.06 | 3.06 | 3.61 |
| Accuracy (%) | 43.82 | 59.83 | 72.45 | 51.70 | 52.32 | 58.14 | 51.52 | 40.66 | 69.02 | 63.73 | 47.70 | 71.54 | 56.87 |

Table 29: ECE (%) of a white-box source-free UDA method SHOT (Liang et al., 2020a) on *Domain-Net*.

| Method | C → S | P → C | P → R | R → C | R → P | R → S | S → P | avg |
|---|---|---|---|---|---|---|---|---|
| No Calib. | 21.57 | 16.14 | 10.03 | 18.18 | 20.86 | 24.71 | 21.52 | 19.00 |
| MatrixScal-src | 27.18 | 19.67 | 12.49 | 19.13 | 16.99 | 21.60 | 20.35 | 19.63 |
| VectorScal-src | 17.79 | 13.95 | 6.46 | 19.31 | 16.25 | 22.17 | 13.20 | 15.59 |
| TempScal-src | 13.91 | 11.32 | 4.81 | 16.76 | 16.47 | 18.99 | 10.63 | 13.27 |
| CPCS | 12.52 | 7.28 | 4.93 | 13.64 | 10.86 | 16.57 | 9.10 | 10.70 |
| TransCal | 16.39 | 23.80 | 25.37 | 24.23 | 18.18 | 15.87 | 14.81 | 19.81 |
| Ensemble | 17.57 | 13.24 | 7.81 | 15.24 | 18.14 | 21.40 | 17.73 | 15.88 |
| PseudoCal | **5.82** | **6.08** | **2.91** | **7.23** | **7.17** | **7.51** | **8.38** | **6.44** |
| Oracle | 2.03 | 3.69 | 1.37 | 2.85 | 2.25 | 2.33 | 2.78 | 2.47 |
| Accuracy (%) | 59.80 | 66.79 | 78.34 | 66.25 | 66.08 | 59.48 | 62.88 | 65.66 |

Table 30: ECE (%) of a black-box source-free UDA method DINE (Liang et al., 2022) on *DomainNet*.

| Method | C → S | P → C | P → R | R → C | R → P | R → S | S → P | avg |
|---|---|---|---|---|---|---|---|---|
| No Calib. | 31.91 | 22.54 | 12.39 | 21.43 | 20.63 | 28.77 | 24.38 | 23.15 |
| Ensemble | 26.38 | 18.72 | 10.83 | 17.03 | 17.53 | 24.28 | 20.18 | 19.28 |
| PseudoCal | **17.86** | **15.12** | **5.30** | **13.71** | **11.14** | **14.44** | **14.75** | **13.19** |
| Oracle | 1.35 | 1.87 | 1.29 | 1.62 | 1.94 | 1.38 | 1.65 | 1.59 |
| Accuracy (%) | 54.26 | 63.00 | 80.69 | 64.52 | 67.13 | 56.75 | 63.81 | 64.31 |

