# OpenReview forum: "Pseudo-Calibration: Improving Predictive Uncertainty Estimation in Domain Adaptation"
_ICLR.cc/2024/Conference — Submitted to ICLR 2024_

### Official Review · Reviewer_M3LQ · 2023-10-15

**Soundness:** 2 fair
**Presentation:** 3 good
**Contribution:** 2 fair
**Rating:** 5
**Confidence:** 4

**Summary:**

The paper proposes a confidence calibration method for unsupervised domain adaptation. First, the adapted network is applied to the samples from the target domain to obtain pseudo labels. Next, a mixup technique is used on pairs of target samples with different pseudo-labels to create a synthetic sample and a soft-label from each pair. Finally, the synthesized pseudo-labeled set is used to find the optimal Temperature Scaling by minimizing the negative likelihood function.

**Strengths:**

The paper is clearly written and easy to follow. The paper reports extensive experiments and shows improved results compared to previous methods. It is interesting to see that previous UDA calibration methods don't work well and in some cases, they are even worse than the uncalibrated model.

**Weaknesses:**

My main concern is that the method is heuristic and not clearly motivated. It is well known that a mixup training procedure yields more calibrated models. However, it is unclear from the paper why learning Temperature Scaling using a convex combination of pseudo-labeled samples yields meaningful temperature values.  How is this method derived from the UDA situation?  What happens if we apply the
same technique in a standard network training setup (without domain shift problems ) to calibrate the network?
The mixture coefficient lambda is set to 0.65.  Again this looks like a heuristic.  is there any theoretical justification for this value?
I would expect a symmetrical mixture combination (lambda=0.5) to be a suitable choice.  The success of a UDA method in different domain shift situations depends on the size of the gap between the domains.  I would expect that the UDA calibration would be dependent on how well the UDA method manages to close the domain gap. I don't see how the proposed method handles this aspect of the UDA calibration problem.

**Questions:**

What happens when we apply the proposed mixup procedure to other UDA calibration methods such as vector scaling and matrix scaling?

What happens if we apply the same technique in a standard network training setup (without domain shift problems ) to calibrate the network?  Does it work well?

---

> ### Author Response · Authors · 2023-11-21
> **Response to Reviewer M3LQ - Part 1**
>
> > **Q1**: My main concern is that the method is heuristic and not clearly motivated.
>
> **A1**: While we acknowledge that our method is heuristic, it's important to highlight that the designs of our PseudoCal method are rooted in clear motivations and can be intuitively explained. **The inspiration for the PseudoCal framework stems from the factorized negative log-likelihood (NLL) objective**. The factorization implies that if two datasets share the same correct-wrong statistics (i.e., an identical count of correct and wrong predictions), they should exhibit similar temperatures after NLL optimization. In the context of PseudoCal, our objective is to synthesize a labeled set that mirrors the correct-wrong statistics of real target samples. We choose to **employ mixup** for sample synthesis due to its simplicity and its **ability to elegantly leverage the target-domain data structure learned by the UDA model**. Additionally, **leveraging the well-established cluster assumption**, we anticipate a sample-level correspondence between mixed samples and real target samples, facilitated by suitable values of the mix ratio $\lambda$. Accurate sample-level correspondences ensure similar correct-wrong statistics, ultimately guaranteeing that the temperature obtained with the labeled pseudo-target set closely approximates the Oracle target-domain temperature.
>
> > **Q2**: It is well known that a mixup training procedure yields more calibrated models. However, it is unclear from the paper why learning Temperature Scaling using a convex combination of pseudo-labeled samples yields meaningful temperature values.
>
> **A2**: We recognize the effectiveness of mixup training in enhancing model calibration performance. However, in the context of UDA, we observe that **this effectiveness falls short of expectations**, as illustrated by the calibration results of **DINE in Table 6**. DINE, a source-free UDA method **utilizing training-stage mixup** with unlabeled target data, exhibits **substantial calibration errors** in the trained model. In contrast, our **PseudoCal significantly reduces ECE errors for DINE models**, as evidenced by the results presented in Table 6.
>
> In **explaining the efficacy of PseudoCal**, it is crucial to highlight the importance of **ensuring similar correct-wrong statistics between the mixed samples and real target samples**. This is a key factor because the NLL objective employed by TempScal is directly influenced by both prediction and label information. The factorization of the NLL implies that correct and wrong predictions (evaluated by labels) have contrasting effects on the NLL optimization process. When there are similar correct-wrong statistics, the NLL optimization employed in TempScal becomes alike, leading to similar temperatures. Note that **for mixed samples**, correctness is assessed based on the predictions of the mixed samples and their corresponding mixed labels. Conversely, **for real target samples**, correctness is evaluated using the target predictions and unavailable target ground truths. **Detailed evidence supporting the similarity** in correct-wrong statistics is provided through **visualizations in Figure 4 in the Appendix** and **quantitative analysis results in Table 10** in the Appendix.
>
> > **Q3**: How is this method derived from the UDA situation? The success of a UDA method in different domain shift situations depends on the size of the gap between the domains. I would expect that the UDA calibration would be dependent on how well the UDA method manages to close the domain gap. I don't see how the proposed method handles this aspect of the UDA calibration problem.
>
> **A3**: **One novelty** of our PseudoCal approach lies in adopting a **target-domain perspective** to address the calibration problem in UDA, distinguishing it from existing methods that primarily rely on source data to address the covariable shift challenge. It's essential to note that **in the UDA field, various studies, including many in the recently popular source-free UDA settings, also operate without source data.**
> Similar to previous target-specific works, PseudoCal **relies on a well-trained UDA model that effectively captures the target-domain data structure.** The absence of source data is compensated by leveraging the knowledge embedded in the UDA model. In situations where the UDA model struggles to learn the target-domain structure, the cluster assumption may not hold, potentially damaging the sample-level correspondence between mixed samples and real target samples. This aspect is evident in **Table 6, where, in the I$\to$S task, both SHOT and DINE models exhibit low accuracy, resulting in high calibration errors** even after being calibrated with PseudoCal. This underscores the **importance of a well-trained UDA model in ensuring the effectiveness of PseudoCal**.

---

> > ### Author Response · Authors · 2023-11-21
> > **Response to Reviewer M3LQ - Part 2**
> >
> > > **Q4**: The mixture coefficient lambda is set to 0.65. Again this looks like a heuristic. is there any theoretical justification for this value? I would expect a symmetrical mixture combination (lambda=0.5) to be a suitable choice.
> >
> > **A4**: First of all, we want to emphasize that **selecting effective hyperparameters remains an open problem** in unsupervised domain adaptation, as demonstrated by a recent large-scale study [1]. It is also worth noting that **competing calibration methods often entail numerous hyperparameters**. For instance, methods like TransCal and CPCS require additional model training, introducing more hyperparameters including model architecture, learning rate, and iterations. In contrast, PseudoCal is a post-hoc method applied to a fixed UDA model, featuring only a single hyperparameter. **Throughout our experiments, which cover diverse UDA methods across various settings, we consistently use a fixed $\lambda$ value of 0.65, highlighting the robustness of PseudoCal and the alleviation of concerns related to hyperparameters.**
> >
> > Our choice of $\lambda=0.65$ is empirically **justified in Appendix D, supported by comprehensive quantitative analysis in Table 10 and qualitative visualizations in Figure 4.** Extremes in $\lambda$ values, whether close to 0.5 or 1.0, demonstrate a noticeable bias towards either wrong or correct predictions, skewing the correct-wrong statistics of the pseudo-target set away from real target samples. Therefore, for a typical UDA model with a mix of correct and wrong predictions, **a moderate $\lambda$ value, such as 0.65 in our case, is recommended.**
> >
> > Regarding the suggestion of **using $\lambda=0.5$**, it's important to note that this is **theoretically unfeasible**. Under this equal-mixing scenario, we **cannot determine the mixed label** for the mixed sample. However, we have considered a closely related case with **$\lambda=0.51$**. The analysis of this case in Appendix D reveals that symmetrical mixtures are a **bad choice**, leading to significant bias towards wrong predictions.
> >
> >
> > > **Q5**: What happens when we apply the proposed mixup procedure to other UDA calibration methods such as vector scaling and matrix scaling?
> >
> > **A5**: Thank the reviewer for the valuable suggestion. We have incorporated the **suggested experiments into our revised paper**, presenting the results in **Table 11 in the Appendix**. The results are summarized below. "No Calib." indicates the UDA model without any calibration, while **source-domain calibration baselines** include "MatrixScal-src," "VectorScal-src," and "TempScal-src."
> >
> > PseudoCal, inspired by the factorized NLL of TempScal, naturally **utilizes TempScal as the default** supervised calibration method for our synthesized labeled pseudo-target set. The critical factor for the success of PseudoCal lies in ensuring that the pseudo-target set exhibits **similar correct-wrong statistics** to the real target samples, which we believe can be generally utilized by different supervised calibration methods. Consequently, we explore the compatibility of PseudoCal with alternative supervised calibration methods such as MatrixScal and VectorScal.
> >
> >
> > |UDA Method|MCD|MCD|BNM|BNM|CDAN|SHOT|PADA|PADA|DINE
> > |:----|:----|:----|:----|:----|:----|:----|:----|:----|:----|
> > |UDA Task|D$\to$A|W$\to$A|Cl$\to$Pr |Pr$\to$Re|R$\to$C|I$\to$S|Ar$\to$Cl|Re$\to$Ar|P$\to$R
> > |No Calib.|16.39|17.03|22.09|15.72|9.83|34.71|20.35|8.31|12.39
> > |MatrixScal-src|17.86|20.28|25.73|15.98|22.11|-|36.55|20.45|-
> > |VectorScal-src|17.75|20.52|16.40|12.36|12.88|-|20.53|9.07|-
> > |TempScal-src|32.09|18.65|15.10|11.64|9.27|-|15.15|6.34|-
> > |PseudoCal(Matrix.)|11.61|13.20|16.07|11.83|15.09|42.86|35.85|27.07|7.65
> > |PseudoCal(Vector.)|11.00|9.32|9.31|6.05|6.37|23.90|5.90|4.19|6.23
> > |PseudoCal(Temp.)|**4.38**|**4.06**|**6.31**|**4.76**|**1.51**|**8.42**|**2.95**|**3.71**|**5.29**
> > |Oracle-target|2.31|1.90|3.14|1.10|1.28|4.39|2.16|2.87|1.29
> > |Accuracy (%)|67.52|66.63|73.69|80.35|52.98|34.29|43.82|63.73|80.69
> >
> > Our observations reveal **two key points**: (i) When a supervised calibration method shows stability and effectiveness using source labeled data for calibration, combining it with our PseudoCal tends to result in a reduced ECE error compared to the no calibration baseline. (ii) Due to the similarity in correct-wrong statistics between the pseudo-target set and real target data, **PseudoCal demonstrates compatibility with both MatrixScal and VectorScal.** However, it consistently achieves the **best calibration performance when paired with TempScal, in line with the findings in [2]** that TempScal generally outperforms MatrixScal and VectorScal.

---

> > > ### Author Response · Authors · 2023-11-21
> > > **Response to Reviewer M3LQ - Part 3**
> > >
> > > > **Q6**: What happens if we apply the same technique in a standard network training setup (without domain shift problems ) to calibrate the network? Does it work well?
> > >
> > > **A6**: **Without domain shifts between the training/validation data and the testing data**, we anticipate that TempScal with a labeled validation set will perform effectively, **eliminating the need to consider any UDA calibration methods**, including ours. We **do not expect PseudoCal to outperform** TempScal under this situation. To address the reviewer’s concern comprehensively, we conduct experiments on the Clipart (C) domain in DomainNet. The dataset is divided into three equal parts: a training set, a validation set, and a testing set. Model training utilizes the training set, and calibration performance is evaluated on the testing set. Four types of calibration results are considered: "No Calib." (model trained on the training set without calibration), "TempScal-val" (model calibrated with labeled validation data), "PseudoCal-test" (model calibrated with unlabeled testing data), and "Oracle-test" (model calibrated with labeled testing data). Our findings indicate that while PseudoCal remains effective in achieving improved calibration performance, it does not exhibit advantages compared to TempScal when using an IID validation set, **aligning with our expectations.**
> > >
> > > |Dataset|Clipart|
> > > |:----|:----|
> > > |No Calib.|17.36|
> > > |TempScal-val|**3.79**|
> > > |PseudoCal-test|3.8|
> > > |Oracle-test|3.16|
> > > |Accuracy (%)|56.65|
> > >
> > >
> > > > **References**:
> > >
> > > [1] Benchmarking validation methods for unsupervised domain adaptation. arXiv preprint arXiv:2208.07360, 2022.
> > >
> > > [2] On calibration of modern neural networks. In International Conference on Machine Learning, 2017.

---

> > > > ### Author Response · Authors · 2023-11-22
> > > > **Thank you for your review**
> > > >
> > > > Dear Reviewer,
> > > >
> > > > We sincerely thank you for dedicating your time and effort to reviewing our paper. We have carefully provided point-to-point responses to your questions or concerns, along with comprehensive revisions, as outlined in the "General response to all reviewers."
> > > >
> > > > Your insights are invaluable to us, and we are open to further discussions before the conclusion of the Author/Reviewer phase. If any aspects of our work remain unclear or if you have additional feedback, we would highly appreciate your input.
> > > >
> > > > Thank you.
> > > >
> > > > Authors of Paper 516

---

### Official Review · Reviewer_SpZE · 2023-10-20

**Soundness:** 2 fair
**Presentation:** 2 fair
**Contribution:** 2 fair
**Rating:** 5
**Confidence:** 4

**Summary:**

The paper proposes a post-hoc calibration method for unsupervised domain adaptation. It utilizes the typical temperature scaling method and adapts it to UDA by generating pseudo-labels on target set, turning the unsupervised setting into a supervised one. Experiments on multiple datasets demonstrate the effectiveness of the method.

**Strengths:**

1. The paper gives the interesting observation about NLL minimization: Temperature scaling provides similar temperatures if two datasets have similar correct-wrong patterns.
2. Experiments are well-done and shows SOTA performance.

**Weaknesses:**

My concerns on this paper mainly concentrates on the proposed method, which I did not fully understand given the limited justifications.
1. Given the first point mentioned in "Strengths", the paper only claimed it as an observation but without further explaination.
2. The method of using Mixup for pseudo-label generation is not convincing to me, even given the "Analysis" section. (a) While the ground-truth labels are not available, can I just assign random labels for each target sample and then adopt PseudoCal (i.e. How is the UDA inference process helpful)? (b) Since $\lambda$ is a fixed scalar and we do not know whether the target sample prediction result is true or not, what is the point of the last 8 lines in "Analysis" part. (c) The pseudo code in Appendix A does not do what equation (3) says for $y_{pt}$. Also, since $\lambda$ is a fixed scalar in (0.5, 1.0), what does the "if else" statement for pseudo_target_y do as "else" is never visited?
3. What is the validation set adopted for Temperature scaling in the experiments?
4. Table 2~5 shows a phenomenon: Temperature scaling sometimes achieves the best performance. What might be the reason?
5. The style of figure 2 is new to me. However, not good-looking nor clear.

**Questions:**

See weaknesses.

**Details Of Ethics Concerns:**

None.

---

> ### Author Response · Authors · 2023-11-21
> **Response to Reviewer SpZE - Part 1**
>
> > **Q1**: Given the first point mentioned in "Strengths", the paper only claimed it as an observation but without further explanation.
>
> **A1**: The observation stems directly from the inherent characteristics of the factorized NLL objective. As clarified in **Section 3.1, the optimization process for correct and wrong predictions in the factorized NLL objective inherently seeks contrasting temperature values.** Specifically, **correct predictions aim for a small temperature to sharpen** the softmax probability distribution, while **wrong predictions aim for a large temperature to flatten** the softmax probability distribution. Consequently, when dividing the confidence ranges of (0.0, 1.0) into multiple small bins and ensuring two datasets share the **same correct-wrong statistics (i.e., an identical count of correct or wrong predictions for each confidence bin)**, a highly consistent NLL optimization process is expected. This consistency results in obtaining similar temperatures after the optimization process. We hope this clarification provides a better understanding of the rationale behind this observation.
>
> > **Q2**: (a) While the ground-truth labels are not available, can I just assign random labels for each target sample and then adopt PseudoCal (i.e. How is the UDA inference process helpful)?
>
> **A2**: PseudoCal is a novel and general framework designed to leverage a synthesized labeled pseudo-target set for optimizing TempScal and obtaining an accurate estimation of the Oracle target-domain temperature. The crucial factor ensuring the success of PseudoCal lies in having this pseudo-target set exhibit similar correct-wrong statistics to the real target samples. **However, we cannot know the actual correct-wrong statistics of the real target samples due to the unavailability of target-domain ground-truth labels.**
>
> Returning to the initial question, while it’s feasible to use randomly assigned labels for calibration with target samples, **there is no assurance of favorable calibration performance because the resulting correct-wrong statistics would be entirely random**. This is demonstrated in **Table 9 under "Pseudo-Label,"** where we employ **hard one-hot target model predictions as labels** for real target samples and apply TempScal, resulting in **substantial calibration errors**.
>
> The effectiveness of PseudoCal with inference-stage mixup relies on establishing a sample-level correspondence between the pseudo-target set and the real target samples, partially guaranteed by the cluster assumption. **The UDA inference process is to obtain predictions** for mixed samples, and **correctness is evaluated** based on these predictions and the corresponding mixed labels. This inference process is critical for **leveraging the cluster assumption** and building the required sample-level correspondence. We have provided comprehensive evidence to illustrate this, including detailed illustrations in **Figure 4 in the Appendix** and extensive quantitative statistics in **Table 10 in the Appendix**.
>
> > **Q3**: (b) Since $\lambda$ is a fixed scalar and we do not know whether the target sample prediction result is true or not, what is the point of the last 8 lines in "Analysis" part.
>
> **A3**: The "Analysis" part in Section 3.2 aims to provide **an intuitive analysis and explanation of how mixed samples can exhibit similar correct-wrong statistics as real target data**. This is crucial for ensuring that applying TempScal with these mixed samples can yield a temperature similar to the Oracle target temperature. In the **revised paper**, we have enhanced the presentation for better clarity, incorporating **more illustrations** of correct-wrong statistics, as demonstrated in **Figure 1(b) and Figure 4 in the Appendix**.
>
> It's essential to clarify that we **analyze the dataset-level similar correct-wrong statistics through the sample-level correspondence of correct or wrong predictions. The definition of correctness differs between mixed samples and real samples.** For mixed samples, correctness is assessed based on the predictions of the mixed samples and their corresponding mixed labels. Conversely, for real target samples, correctness is evaluated using the target predictions and unavailable target ground truths. PseudoCal never accesses target ground truths and relies solely on target images and their model predictions to synthesize a labeled pseudo-target set, comprising mixed samples and mixed labels. **For a quantitative analysis of the sample-level correspondence, we kindly refer the reviewer to Appendix D.**
>
> The use of **target ground truths serves exclusively for analysis and explanation purposes** to study the success of PseudoCal. This involves explaining and showcasing similar correct-wrong statistics between the pseudo-target set and real target samples, as discussed in both the Analysis part in Section 3.2 and Appendix D.

---

> > ### Author Response · Authors · 2023-11-21
> > **Response to Reviewer SpZE - Part 2**
> >
> > > **Q4**: (c) The pseudo code in Appendix A does not do what equation (3) says for $y_{pt}$. Also, since $\lambda$ is a fixed scalar in (0.5, 1.0), what does the "if else" statement for pseudo_target_y do as "else" is never visited?
> >
> > **A4**: We appreciate the reviewer's attention to detail. Regarding the concern about the pseudocode, we want to emphasize that **our pseudocode accurately represents the implementation of PseudoCal**, and there is **no inconsistency between the code and the paper**. While **Equation 3** provides a **general formulation of mixup** without specifying whether $y_t$ represents hard one-hot pseudo-labels or soft predictions, we clarify in **"Impact of mix ratio $\lambda$" in Section 4.3** that we use hard one-hot pseudo-labels for $y_t$, **aligning with our pseudocode**.
> > Additionally, we would like to clarify that we limit the discussion of $\lambda$ to the range of (0.5, 1.0) for simplicity and clarity in our submission because the ranges of (0.0, 0.5) and (0.5, 1.0) are **symmetric** situations. Regarding the **"if else"** statement in the pseudocode, it is important to note that the "else" branch is not typically executed due to the restricted range of $\lambda$ (0.5 to 1.0). Nevertheless, this structure is included **for comprehensive logical coverage**, aiming to **avoid potential bugs** that may arise if the user is not aware of the predefined range for $\lambda$.
> >
> > > **Q5**: What is the validation set adopted for Temperature scaling in the experiments?
> >
> > **A5**:
> > - **For the baseline “TempScal-src,”** representing source-domain calibration, we use **a labeled source validation set** to calibrate the UDA model.
> > - **For the baseline “Oracle,”** we apply TempScal using model **predictions of unlabeled target data** and the corresponding **target ground truths** for calibration. This serves as an upper bound for target-domain calibration performance as a reference point.
> > - **For our PseudoCal**, we synthesize a labeled pseudo-target set comprising mixed samples and mixed labels. Subsequently, we **use the UDA model to infer predictions for these mixed samples** and apply TempScal using these predictions and **mixed labels** for calibration.
> >
> > > **Q6**: Table 2~5 shows a phenomenon: Temperature scaling sometimes achieves the best performance. What might be the reason?
> >
> > **A6**: The baseline **“TempScal”** in our initial submission **has been replaced by “TempScal-src,”** explicitly indicating that it is **a source-domain calibration baseline using a labeled source validation set**. The occasional occurrence of “TempScal-src” achieving the best performance is **attributed to the labeled source validation set coincidentally having similar correct-wrong statistics to the unlabeled testing target data.** We want to emphasize that this infrequent phenomenon **does not impact our contributions**. In the **majority of tasks**, as demonstrated in our experiments, **“TempScal-src” shows only marginal calibration improvement** compared to other UDA calibration methods, including our PseudoCal. However, practitioners and researchers in the domain adaptation community are in pursuit of **reliable and consistently effective calibration methods across various UDA tasks and settings**. Evidently, **"TempScal-src" does not align** with this criterion.
> >
> > > **Q7**: The style of figure 2 is new to me. However, not good-looking nor clear.
> >
> > **A7**: We apologize if Figure 2 appears unclear or less aesthetically pleasing. We would appreciate it if the reviewer could provide specific feedback on areas that require improvement, allowing us to enhance the figure for better clarity and visual appeal.

---

> > > ### Comment · Reviewer_SpZE · 2023-11-21
> > > **Thank you for your response**
> > >
> > > The authors have fully addressed most of my concerns and revised the paper to a large degree. I am willing to raise my score given the effort the authors have made.
> > >
> > > For Figure 2, it is just not academia enough to me... However, it is only my personal preference and I won't treat it as a weakness if the other reviewers are OK with it. This is also my attitude towards the paper, as I won't stand in the way if all other reviewers agree to accept it.

---

> > > > ### Author Response · Authors · 2023-11-21
> > > > **Thank you for increasing the score**
> > > >
> > > > We sincerely appreciate the reviewer's acknowledgment that we have "fully addressed most of the concerns and revised the paper to a large degree".
> > > >
> > > > We also extend our respect for the reviewer's personal preference regarding our paper. Regardless of the final decision, whether it be acceptance or rejection, we remain committed to thoroughly addressing all questions raised by the reviewer and other reviewers. We believe that this constructive feedback is invaluable and can significantly enhance the quality of our paper. Thank you.

---

### Official Review · Reviewer_dbpq · 2023-10-31

**Soundness:** 3 good
**Presentation:** 3 good
**Contribution:** 2 fair
**Rating:** 6
**Confidence:** 4

**Summary:**

In this work, the authors propose PseudoCal, which applies a mix-up strategy to generate pseudo-labels for unlabelled data to predict calibration uncertainty in an unsupervised domain adaptation problem. The proposed approach is examined on multiple image classification and semantic segmentation benchmarks, showing its superiority over SOTA.

**Strengths:**

+ The paper is written well, and the idea is easy to follow.
+ The experiments cover extensive benchmarks of multiple settings, e.g closed-set, partial-set and Source-free,  and show marginal performance boosing w.r.t ECE.
+ A simple and clear pseudo code is provided in the appendix.

**Weaknesses:**

- The novelty is weak because mix-up is a well-known strategy in image classification tasks. From my perspective, the methodology is the same as the previous ICLR 2018 work, "mixup: BEYOND EMPIRICAL RISK MINIMIZATION".
- The mix-up is conducted at input level, which would introduce artifacts. It would be good to include some visualization in the appendix together with quantitative results.
- I am skeptical about some numbers in the table. When lamba = 0 or 1, PseudoCal becomes typical TempScal with pure negative or positive samples. However, the ECEs shown in the tables are different. For example, in Home dataset, TempScal gets 11.06 but PseudoCal with lamba = 1 achieves ~20.
- How did the author get the results of other methods, cite from the original papers, or re-implement those approaches?

**Questions:**

Please see the weakness section.

---

> ### Author Response · Authors · 2023-11-20
> **Response to Reviewer dbpq - Part 1**
>
> > **Q1**: The novelty is weak because mix-up is a well-known strategy in image classification tasks. From my perspective, the methodology is the same as the previous ICLR 2018 work, "mixup: BEYOND EMPIRICAL RISK MINIMIZATION".
>
> **A1**: We appreciate the reviewer's comments and **respectfully disagree** with the assertion that **“the methodology is the same as the previous ICLR 2018 work [1]”.**
>
> 1) While acknowledging the foundational work of [1] and its impact on the field, we **highlight distinct differences** in how mixup is employed in our work, as **discussed in Section 4.3 "Comparison with training-stage mixup"** and summarized below:
>
> - **Different application stage**: Unlike existing mixup works [1, 2], which leverage mixup during the training stage for model training, **our PseudoCal** approach employs mixup during the **inference stage**, specifically for UDA model calibration, **without altering model parameters**.
>
> - **Different mix ratio ($\lambda$)**: In prior works **[1, 2], effective** mixup commonly utilizes **$\lambda$ values near 1**, typically **randomly sampled** from a beta-distribution, i.e., $\lambda \sim Beta(\alpha, \alpha)$, where $\alpha \in [0.1, 0.4]$. Notably, $\lambda$ values significantly **deviating from 1.0** are considered **ineffective due to the manifold intrusion problem [2, 3]**. On the contrary, **PseudoCal** exhibits **optimal** performance with **fixed $\lambda$** values significantly **deviating from 1.0**. However, employing **$\lambda=1.0$** in PseudoCal (i.e., PseudoCal($\lambda=1.0$)) leads to **significant calibration errors**.
>
> - **Different calibration performance**: We present compelling evidence from two perspectives. **First**, we examine **DINE [4]**, a source-free domain adaptation method **employing the training-stage mixup** strategy [1, 2] during model training. As outlined in **Table 6**, our observations indicate that UDA models trained with DINE still manifest substantial calibration errors. In contrast, our **PseudoCal effectively calibrates the DINE model**, leading to notable reductions in calibration errors. Specifically, the ECE (%) error drops from 21.81 to 12.82 on DomainNet and from 58.85 to 47.76 on ImageNet. Furthermore, when we **substitute our mixup strategy with the training-stage mixup [1, 2] in our PseudoCal framework**, denoting this variant as **"Mixup-Beta" in Table 9**, we observe the superior performance of our mixup strategy. These significant performance differences underscore the distinct nature and advantage of our mixup strategy.
>
> 2) Moreover, it is crucial to highlight that our usage of mixup serves as only one technical facet of our framework; **the novelty of our paper extends far beyond this component**, as summarized below.
>
> - **Novel target-domain perspective**: We approach the challenging calibration problem in UDA as **an unsupervised calibration issue in the target domain**, **diverging from the previous** treatment as a **cross-domain covariate shift** problem seen in competing approaches listed in Table 1 of our submission.
>
> - **Novel TempScal objective factorization**: We introduce a **unique factorization of the negative log-likelihood (NLL)** objective used in TempScal, a distinctive feature not identified in relevant UDA model calibration studies compared in Table 1.
>
> - **Novel PseudoCal framework**: Our innovative post-hoc framework, PseudoCal, addresses model calibration in domain adaptation through a novel strategy for synthesizing pseudo-target samples. This sets it apart from competing methods, as thoroughly demonstrated in **Table 1 of our submission**.
>
> - **Comprehensive empirical study**: Our extensive experiments with PseudoCal encompass 10 domain adaptation methods across 5 UDA scenarios. **Notably, we include the calibration under source-free UDA settings and domain-adaptive semantic segmentation for the first time**. PseudoCal consistently exhibits significant outperformance over all competing methods.
>
> >**Q2**: The mix-up is conducted at input level, which would introduce artifacts. It would be good to include some visualization in the appendix together with quantitative results.
>
> **A2**: Thank you for your valuable suggestion. We have **added visualizations** of input-level mixup for all UDA benchmarks in **Figure 5 of the Appendix** for a comprehensive understanding.

---

> > ### Author Response · Authors · 2023-11-20
> > **Response to Reviewer dbpq - Part 2**
> >
> > >**Q3**: I am skeptical about some numbers in the table. When lamba = 0 or 1, PseudoCal becomes typical TempScal with pure negative or positive samples. However, the ECEs shown in the tables are different. For example, in Home dataset, TempScal gets 11.06 but PseudoCal with lamba = 1 achieves ~20.
> >
> > **A3**: Regarding the $\lambda$ value, we limited the discussion to the range of (0.5, 1.0) for simplicity and clarity in our submission because the ranges of **(0.0, 0.5) and (0.5, 1.0) are symmetric** situations. Therefore, when we **set $\lambda = 1.0$**, it means removing the mixup operation from PseudoCal, and PseudoCal essentially becomes the **TempScal calibration using predictions of unlabeled target data and the model-predicted (hard one-hot) pseudo-labels of these data. The entry "PseudoCal ($\lambda=1.0$)" in Table 10 (Appendix) is the same as the baseline "Pseudo-Label" in Table 9.**
> > We want to clarify that in our initial submission, the "TempScal" rows in all tables represented the source-domain calibration baseline that uses a labeled source validation set for calibration. Therefore, **"TempScal" is different from "PseudoCal ($\lambda=1.0$)." To avoid confusion, in the revision, we have replaced "TempScal" with "TempScal-src"** and explicitly clarified in **Section 2 and "Calibration baselines" in Section 4.1** that "TempScal-src" denotes calibration using only labeled source data, which is fundamentally different from other calibration methods.
> >
> > >**Q4**: How did the author get the results of other methods, cite from the original papers, or re-implement those approaches?
> >
> > **A4**: Indeed, we meticulously **re-implemented all calibration baselines and UDA methods** featured in our experiments. The UDA methods were trained using their officially released code. As for **calibration methods, we employed the codebase of TransCal [5], accessible at https://github.com/thuml/TransCal**. For each task, we applied all calibration baselines to calibrate the same UDA model. Our experiments underwent five repetitions, and the averaged results are reported in the tables. Our work stands out for providing comprehensive empirical comparisons among all calibration baselines in UDA settings. **Notably, we ventured into evaluating domain adaptive semantic segmentation tasks and source-free UDA settings for the first time.** In these new settings, some existing baselines could not be applied due to their focus on classification tasks and the utilization of source data. Therefore, our comparisons are limited to applicable calibration methods such as Ensemble.
> >
> > > **References**:
> >
> > [1] mixup: Beyond empirical risk minimization. In International Conference on Learning Representations, 2018.
> >
> > [2] On mixup training: Improved calibration and predictive uncertainty for deep neural networks. In Advances in Neural Information Processing Systems, 2019.
> >
> > [3] Mixup as locally linear out-of-manifold regularization. In AAAI Conference on Artificial Intelligence, 2019.
> >
> > [4] Domain adaptation from single and multiple black-box predictors. In IEEE Conference on Computer Vision and Pattern Recognition, 2022.
> >
> > [5] Transferable calibration with lower bias and variance in domain adaptation. In Advances in Neural Information Processing Systems, 2020.

---

> > > ### Author Response · Authors · 2023-11-22
> > > **Thank you for your review**
> > >
> > > Dear Reviewer,
> > >
> > > We sincerely thank you for dedicating your time and effort to reviewing our paper. We have carefully provided point-to-point responses to your questions or concerns, along with comprehensive revisions, as outlined in the "General response to all reviewers."
> > >
> > > Your insights are invaluable to us, and we are open to further discussions before the conclusion of the Author/Reviewer phase. If any aspects of our work remain unclear or if you have additional feedback, we would highly appreciate your input.
> > >
> > > Thank you.
> > >
> > > Authors of Paper 516

---

### Official Review · Reviewer_fgV1 · 2023-11-02

**Soundness:** 2 fair
**Presentation:** 3 good
**Contribution:** 2 fair
**Rating:** 6
**Confidence:** 4

**Summary:**

This paper addresses the unsupervised domain adaptation from the aspect of target-domain specific unsupervised. Specifically, the PseudoCal framework is proposed. In the inference stage, this method uses a mixup strategy to generate a pseudo-target set and perform supervised calibration on it. To clarify the effectiveness of the PseudoCal, the authors conduct analysis through the cluster assumption. The experiments show that the proposed method outperforms other state-of-the-art baselines on several datasets.

**Strengths:**

1.	The paper presents a new perspective to make up the UDA uncertainty calibration research and unifies the covariate shift and label shift scenarios in UDA.
2.	The proposed method uses the generated pseudo-target set to convert the unsupervised problem into a supervised one.
3.	Good set of experiments for yielding state-of-the-art results on several datasets.

**Weaknesses:**

1.	The analysis of the mixup in the inference strategy is not sufficient. The comparison between the training stage mixup strategy in other methods and the inference stage is not obvious, the authors should strengthen your highlight.
2.	Writing can be improved. In addition, the experiment results could present more in the “Accuracy” metric, which is also intuitive.

**Questions:**

How many pseudo-target samples will be generated? Does the number of n_{pt} be the same with target sample number?
	To find the value of mix ratio \lambda, the authors analyze the ECE on different UDA scenarios. However, it seems that the black-box source-free scenario does not have a corresponding analysis.
	The mixup strategy and the temperature scaling both are calibration methods. What are the results if the pseudo target set is just generated by the model predicting, not using the mixup strategy?

---

> ### Author Response · Authors · 2023-11-20
> **Response to Reviewer fgV1 - Part 1**
>
> > **Q1**: The analysis of the mixup in the inference strategy is not sufficient. The comparison between the training stage mixup strategy in other methods and the inference stage is not obvious, the authors should strengthen your highlight.
>
> **A1**: In Section 4.3, we delve into a detailed analysis of the mixup operation's impact on the calibration performance of our PseudoCal. **Two key implementation-level factors are considered**: the use of hard one-hot labels or soft predictions for the mixup process in Equation 3, and the value of the mix ratio $\lambda$. Empirical results across various UDA methods and settings are presented in **Figure 3(c)-(d)**. Specifically, we highlight the crucial role of $\lambda$ values, **emphasizing in the main text that "a $\lambda$ value closer to 0.5 generates more ambiguous samples, resulting in increased wrong predictions, whereas a $\lambda$ value closer to 1.0 has the opposite effect." For readers interested in detailed quantitative analysis, we provide a comprehensive discussion in Appendix D.**
>
> In comparing our approach to mainstream training-stage mixup [1,2], our goal is to **highlight the novel aspects of our mixup strategy**. We emphasize three key novelties. **Firstly**, we employ **mixup during the inference stage** rather than the traditional training stage. **Secondly**, our mixup strategy exhibits effectiveness with **a fixed $\lambda$ value significantly deviating from 1.0**, a departure from the prevalent use of values close to 1.0 in existing mixup approaches. **Thirdly**, applying training-stage mixup directly to train a UDA model, as seen in UDA methods like **DINE, still results in substantial calibration errors.** Conversely, our PseudoCal can significantly reduce ECE errors for such models, as evidenced by the DINE results presented in **Table 6**. Additionally, in the ablation study of PseudoCal, where we **replace our $\lambda$ usage** with values generated through the beta-distribution employed in existing mixup, the results under **"Mixup-Beta" in Table 9** show that our mixup strategy outperforms traditional mixup.
>
>
> > **Q2**: Writing can be improved. In addition, the experiment results could present more in the “Accuracy” metric, which is also intuitive.
>
> **A2**: Thank you for your valuable feedback on writing. We have revised the manuscript to enhance clarity and provide more detailed results. Regarding the "Accuracy" metric, we have clarified in the **first paragraph of Section 4.2** that, in all tables, "‘Accuracy’ (%) denotes the target accuracy of the **fixed UDA model**." This addition only serves to offer readers a clear understanding of the UDA model's discriminative ability in the target domain. **As indicated in Table 1 and Figure 2, our PseudoCal** method is a **post-hoc** calibration approach that solely **utilizes the UDA model for inference without changing "Accuracy" of the model**. It's worth noting that many existing calibration methods, such as **TempScal, CPCS, and TransCal**, are **also post-hoc** in nature.
>
> > **Q3**: How many pseudo-target samples will be generated? Does the number of $n_{pt}$ be the same with target sample number?
>
> **A3**: Yes, for all experiments, the number of $n_{pt}$ is almost the same as the target sample number $n_t$. Specifically, in the **Implementation details**, we mentioned that “We use the UDA model for **one-epoch inference with mixup to generate the labeled pseudo-target set**.” Additinoally, as indicted in our pseudocode in **Appendix A**, during the inference process, we **only mix real target samples that have different pseudo labels**, i.e., cross-cluster perturbation. Therefore, in general, **$n_{pt}$ would be slightly smaller than $n_t$** due to **ignoring the pairs of the shuffled real target samples that have the same pseudo label**.
>
> > **Q4**: To find the value of mix ratio \lambda, the authors analyze the ECE on different UDA scenarios. However, it seems that the black-box source-free scenario does not have a corresponding analysis.
>
> **A4**: We appreciate the thorough examination of our ablation experiments in Table 9 and analysis experiments in Table 10 (Appendix). Both tables maintain a consistent setting for UDA methods. In **our initial submission**, we presented the black-box source-free method **DINE** and the white-box source-free method **SHOT under the same parent UDA setting, i.e., source-free UDA**. Consequently, we only reported SHOT ablation results. **Following the valuable suggestion, we have included the results of DINE in both Table 9 and Table 10** to facilitate comprehensive comparisons **without altering our initial conclusions**.

---

> > ### Author Response · Authors · 2023-11-20
> > **Response to Reviewer fgV1 - Part 2**
> >
> > >**Q5**: The mixup strategy and the temperature scaling both are calibration methods. What are the results if the pseudo target set is just generated by the model predicting, not using the mixup strategy?
> >
> > **A5**: Yes, the mixup strategy has been employed during the model training stage [1, 2] to enhance calibration performance. Temperature scaling is a well-known supervised calibration method effective for testing data that are IID (independent and identically distributed) to the training data. **However, we observed that both methods struggle to effectively address the calibration problem in the UDA setting**, where testing data in the target domain is entirely unlabeled, and severe distribution shifts exist between the labeled source data used in training and the target data. In all our tables, **"TempScal-src" denotes calibrating the UDA model with a labeled source validation set**, showing only marginal reduction in the ECE error compared to our PseudoCal. Additionally, in **Table 6, the UDA method DINE employs the mixup strategy in model training**, and the **"No Calib."** results reveal that the trained model still exhibits **large ECE errors** in the target domain.
> >
> > Regarding the baseline that **directly uses predictions with unlabeled target data and model-predicted pseudo-labels for calibration (without the mixup strategy or using $\lambda$ of 1.0)**, we have included it in **Table 9 as "Pseudo-Label" and "Filtered-PL", and in Table 10 (Appendix) as “PseudoCal ($\lambda=1.0$).”** The results demonstrate that directly using model-predicted pseudo-labels tends to **be biased toward all correct predictions**, leading to a large calibration error. Concerning this extreme case, i.e., **PseudoCal ($\lambda=1.0$)**, we offer detailed quantitative evidence in **Table 10**, complemented by visualizations illustrating correct-wrong statistics in **Figure 4**, both available in Appendix D. We have discussed this baseline in the **"Ablation study on pseudo-target synthesis" in Section 4.3.**
> >
> >
> > > **References**:
> >
> > [1] mixup: Beyond empirical risk minimization. In International Conference on Learning Representations, 2018.
> >
> > [2] On mixup training: Improved calibration and predictive uncertainty for deep neural networks. In Advances in Neural Information Processing Systems, 2019.

---

> > > ### Author Response · Authors · 2023-11-22
> > > **Thank you for your review**
> > >
> > > Dear Reviewer,
> > >
> > > We sincerely thank you for dedicating your time and effort to reviewing our paper. We have carefully provided point-to-point responses to your questions or concerns, along with comprehensive revisions, as outlined in the "General response to all reviewers."
> > >
> > > Your insights are invaluable to us, and we are open to further discussions before the conclusion of the Author/Reviewer phase. If any aspects of our work remain unclear or if you have additional feedback, we would highly appreciate your input.
> > >
> > >
> > > Thank you.
> > >
> > > Authors of Paper 516

---

### Author Response · Authors · 2023-11-20
**General response to all reviewers**

We express our gratitude to all reviewers for providing valuable feedback. In our responses to each reviewer, we reply to questions or comments point by point. The manuscript has been revised accordingly, and the **main changes are highlighted in blue**.



1. In **Section 1**, we have introduced **a visual illustration of the correct-wrong statistics** for both the synthesized pseudo-target set and real target set **in Figure 1(b)** to enhance clarity. Additionally, we have refined the presentation of the introduction of our method, PseudoCal.

2. In **Sections 2 and 4.1**, we have made it explicit that we **have replaced the original "TempScal" with "TempScal-src"** to prevent any potential misunderstanding. This **source-domain calibration baseline** involves **using a labeled source validation set** to calibrate the UDA model, followed by evaluating this source-calibrated model on the target domain.

3. In **Section 3.1**, we have clarified our analysis of the **"Oracle" calibration** case on the target domain. This involves **applying TempScal with unlabeled target data and unattainable target ground truths**. Through factorizing the NLL objective in this optimization process, we uncover the insight with respect to correct-wrong statistics.

4. In **Section 3.2 and Appendix D**, we have elucidated the relationship between **dataset-level correct-wrong statistics and sample-level correspondence**. To mitigate potential confusion or misunderstandings, we have enhanced our presentation of the analysis. Additional details, including **results in Table 10, visualizations of correct-wrong statistics in Figure 4, and illustrations of mixed images in Figure 5**, have been incorporated.

5. In **Section 4.3**, we have clarified **the novelties in our use of mixup compared to existing training-time mixup works**. We emphasize differences **in applied stages, mix ratios, and calibration performance.** Furthermore, we explicitly state that PseudoCal without mixup (**setting $\lambda$ to 1.0**) is **equivalent** to directly using **pseudo-labeled target samples for TempScal**, as demonstrated in **Table 9**.

6. In **Appendix A**, we have included additional comments about our code and explained the use of the **if-else statement**.

7. In **Appendix F**, we have replaced the notations "MatrixScal" and "VectorScal" with **"MatrixScal-src" and "VectorScal-src"** to signify that these results are obtained after **supervised calibration on source labeled data**. Additionally, in **Table 11**, we have incorporated detailed results of **combining PseudoCal with** supervised calibration methods other than TempScal, specifically **MatrixScal and VectorScal**.

We sincerely hope that **our work will draw increased attention** to the significant yet under-explored calibration problem in the domain adaptation community. We believe **our simple yet effective PseudoCal method can offer novel insights for future practical calibration solutions**. We appreciate the valuable feedback from the reviewers, and we have diligently addressed their concerns in the revised manuscript. We remain **open to addressing any further questions or concerns before the end of the Author/Reviewer phase** and appreciate the time and consideration.

---

### Meta-Review · Area_Chair_j1v1 · 2023-12-07

**Metareview:**

This paper studies the uncertainty calibration problem in the unsupervised domain adaptation (UDA) setting. The main idea is to use pseudo-labels generated from UDA to help supervise the post hoc calibration process. Extensive experiments are conducted on various UDA datasets and methods.

Strengths: all reviewers agree that the idea is easy to follow and the experiments are very solid.

Weaknesses: the major concern is the novelty and the comparison/relation to the prior work that utilizes a similar strategy. Given the proposed method stems from empirical observation and the usage of pseudo-labels is a very common strategy in the domain adaptation field, the paper may benefit from developing a theoretical understanding and adding more discussion about the comparison with the related work.

**Justification For Why Not Higher Score:**

The paper has a lot of merits. However, given the simple idea, the paper may benefit from incorporating novel theoretical analysis to enhance contribution.

**Justification For Why Not Lower Score:**

N/A

---

### Decision · Program_Chairs · 2024-01-16

Reject